# Continuum Models of Membrane Fusion: Evolution of the Theory

**DOI:** 10.3390/ijms21113875

**Published:** 2020-05-29

**Authors:** Sergey A. Akimov, Rodion J. Molotkovsky, Peter I. Kuzmin, Timur R. Galimzyanov, Oleg V. Batishchev

**Affiliations:** Laboratory of Bioelectrochemistry, A.N. Frumkin Institute of Physical Chemistry and Electrochemistry, Russian Academy of Sciences, 31/4 Leninskiy Prospekt, 119071 Moscow, Russia; swinka87@gmail.com (R.J.M.); copper1956@mail.ru (P.I.K.); gal_timur@yahoo.com (T.R.G.); olegbati@gmail.com (O.V.B.)

**Keywords:** membrane fusion, lipid membranes, theory of elasticity, stalk, leaky intermediates, fusion proteins, hydrophobic interactions, hydration pressure, pore formation

## Abstract

Starting from fertilization, through tissue growth, hormone secretion, synaptic transmission, and sometimes morbid events of carcinogenesis and viral infections, membrane fusion regulates the whole life of high organisms. Despite that, a lot of fusion processes still lack well-established models and even a list of main actors. A merger of membranes requires their topological rearrangements controlled by elastic properties of a lipid bilayer. That is why continuum models based on theories of membrane elasticity are actively applied for the construction of physical models of membrane fusion. Started from the view on the membrane as a structureless film with postulated geometry of fusion intermediates, they developed along with experimental and computational techniques to a powerful tool for prediction of the whole process with molecular accuracy. In the present review, focusing on fusion processes occurring in eukaryotic cells, we scrutinize the history of these models, their evolution and complication, as well as open questions and remaining theoretical problems. We show that modern approaches in this field allow continuum models of membrane fusion to stand shoulder to shoulder with molecular dynamics simulations, and provide the deepest understanding of this process in multiple biological systems.

## 1. Introduction

Cells are open nonequilibrium systems. In the process of their life, cells constantly receive from outside and secrete from inside various low- and macromolecular substances. For normal homeostasis, a cell needs to maintain certain chemical and electrochemical gradients of various substances between the cytoplasm and the extracellular environment. Cellular membranes play a role of fine-tunable barriers between the cell interior and the environment. Their basic element is a bilayer of amphiphilic lipid molecules [1]. The amphiphilic nature of lipids determines extremely low permeability of membranes for various substances: polar or charged molecules cannot overcome the hydrophobic region of the membrane, and hydrophobic ones entering the membrane cannot leave it [2,3]. As a result, the passive transport of substances through the lipid bilayer is hampered, and it is managed by the cell via the work of certain channel proteins. Violation of the barrier function of the membranes is deadly for any living cell. Nevertheless, rearrangements of the membranes constantly occur in the eukaryotic cell. They are coupled with processes of consumption or the release of macromolecules and large objects by the membranes during endo- and exocytosis, renewal, and recycling of the membranes themselves, and cell–cell interactions [4,5]. One of the most important membrane processes is a fusion of the membranes, which includes their merger and further combining of water volumes confined by them. One characteristic example of the fusion in eukaryotic cells is synaptic transmission; its key stage is the fusion of synaptic vesicles containing neurotransmitters with a presynaptic plasma membrane [6]. In addition to the fusion of subcellular structures (e.g., vesicles and organelles), the fusion also occurs between the cells themselves, for example, in the processes of fertilization, carcinogenesis, and muscle tissue growth [7,8,9,10]. For instance, the skeletal muscles consist of bundles of elongated multinucleated myofibrils formed by the fusion of mononuclear myoblasts. Myoblast fusion is essential for the maintenance, growth, and regeneration of myofibrils [11]. Membrane fusion is an important stage in the life cycle of enveloped viruses, which include such dangerous pathogens as the human immunodeficiency virus (HIV), the coronaviruses SARS-CoV, SARS-CoV-2, MERS-CoV, hepatitis C, the Ebola virus, and influenza [12,13,14]. To infect a cell, a virus has to incorporate its genetic material into the cytoplasm. Initially, it is separated from the cytoplasm of the cell by two membranes, viral and cellular ones. As a result of the fusion of these membranes, the water volumes enclosed by them unite, i.e., the cell cytoplasm and the inner space of the virion containing viral genetic material become continuously connected. Some enveloped viruses, such as HIV, merge directly with the plasma membrane of an infected cell; others, such as the influenza virus, enter a cell during endocytosis and fuse with the endosomal membrane inside the cell [15].

Certain proteins called fusion proteins govern the fusion of membranes [16,17]. For example, in the secretion of hormones and neurotransmitters, the complex of SNARE proteins catalyzes membrane fusion [18,19]. Intercellular fusion is also provided by specific proteins, which, however, are identified only in rare cases and remain unknown for most cases [20]. Conformational changes of the fusion proteins of enveloped viruses leading to the merger of membranes are usually triggered by binding to a specific receptor on the cell surface, or by exposure to the acidic environment of late endosomes [13,21]. The fusion proteins are aimed to deform interacting membranes with their subsequent merger and mixing of initially separated water volumes. The work these proteins must perform to complete the fusion process in each case determines their structure, quantity, the presence of auxiliary proteins, etc.

Experimental studies of the fusion process are difficult, due to the extremely small volume of fusion intermediate structures which characteristic dimensions are approximately 10 × 10 × 10 nm^3^ [22,23]. The contact area of two merging membranes may vary in a wide range, e.g., from 50–650 nm^2^ for the contact of a synaptic vesicle with presynaptic membrane [24], to several square microns for the fusion of two cells [25]; however, the characteristic lateral size of fusion intermediate structures is of the order of several membrane thicknesses, i.e., ~10 nm [23,25,26]. In addition, a characteristic time of membrane fusion varies in the range of several orders of magnitude [27]: for example, the fusion during synaptic transmission occurs in about tens to hundreds of microseconds [28,29]; about a few minutes is required for the fusion of the cell membrane and the membrane of enveloped viruses [30,31]; it takes about several days for intercellular fusion to occur [32]. A study of the fast synaptic fusion is difficult, due to a large number of participating proteins [33,34,35] aimed at ensuring a high speed of the process. Research of the slow intercellular fusion is hindered, because the acquisition of statistically reliable data takes a large amount of time. The characteristic time of fusion induced by enveloped viruses is several minutes, which is convenient both for registration (devices with high time resolution are not required) and for sufficient experimental data collection. Additionally, as a rule, only one or two proteins are responsible for fusion in viruses; in particular, the influenza A virus has only one fusion protein—hemagglutinin [13]. In this regard, virus-induced fusion, and, in particular, the fusion of influenza virions with model and cellular membranes, is currently widely used as a convenient model system for detecting patterns of membrane fusion mediated by fusion proteins [36]. Similar intermediate structures (e.g., hemifusion diaphragm) are observed for fusion processes induced by enveloped viruses and during exocytosis in secreting cells. The relatively slow fusion provided by the *trans*-SNARE complex reconstituted in model membranes [37] seems to qualitatively resemble the viral-induced fusion. However, the analogy with viral fusion could be misleading if directly transmitted to the ultrafast fusion in vivo nerve terminals. Protein machines catalyzing the fusion of enveloped viruses with target cells and synaptic vesicles with presynaptic membrane are intended to provide very different quantitative characteristics of the fusion—first of all, different rates of fusion. Thus, there is no reason to assume similar mechanisms and trajectories of the fusion process in these two cases. The validity of the enveloped virus model for the fusion process should be explicitly justified in each particular case.

To generalize and systematize data obtained from experiments on virus-induced fusion, to identify its main patterns and apply them to other types of fusion mediated by specialized proteins, one should build detailed physical models of the process. In these models, both the work of proteins and the physicochemical processes undergone by lipid bilayers should be taken into account. The main criterion for the validity of a constructed model is its energy feasibility. For the relatively simple virus-induced fusion, the number of proteins involved in each fusion event is approximately known [21,38]. The mechanical energy stored in each fusion protein can also be estimated [39]. In the process of fusion, lipid bilayers of biological membranes inevitably deform [23], and the deformational energy is usually so high that it is that mainly determines the speed of the whole process. For the feasibility of the fusion process, the energy stored in the fusion proteins should be sufficient to deform lipid bilayers during the process.

The energy of the membrane deformations can be calculated in the framework of an appropriate theory of elasticity. Traditionally, the theory of elasticity of lipid membranes is based on the similarity of lipid bilayers with smectic A liquid crystals, which have layered structure as well [40]. This similarity is complemented by strong anisotropy of lipid molecules, the longitudinal size of which significantly exceeds their lateral sizes, as well as by the presence of rather rigid parts in the structure of lipids (glycerol in the case of glycerolipids, sphingosine in the case of sphingolipids, etc.) and dipole moments in the polar parts of molecules. A state of the deformed smectic is determined by the spatial distribution of a director **n**, which is a unit vector characterizing the average orientation of molecules at a given point of the membrane. In the approximation of small deformations (Hooke’s law), only splay deformation can occur in smectics. This deformation is described by the divergence of the director, div(**n**) [40]. The energy of a deformed smectic A is written as follows [40]:(1)$$W_{s}=\int \frac{k_{11}}{2}{\left[\mathrm{div}\left(n\right)\right]}^{2}dV,$$
where *k*_11_ is the splay modulus; the integration is over the volume of the smectic A sample. Since the thickness of membranes has a molecular scale (~4 nm), and usually it is much smaller than the characteristic lateral size of the membrane, it is convenient to integrate the deformation energy Equation (1) over the membrane thickness. This allows assigning the director distribution not to space, but some reference surface located inside the membrane:(2)$$W_{b}=\int \frac{B}{2}{\left[\mathrm{div}\left(n\right)\right]}^{2}dS.$$
where *W_b_* is the energy of the deformed lipid bilayer; *B* is the splay modulus of the membrane related to the reference surface; the integration is over the reference surface area. In a bulk liquid crystal, the splay of smectic A layers is accompanied by the change in their shape, so that the normal to the surface of the layer always coincides with the director [40]. Within the framework of the firstly developed classical theory of elasticity of lipid membranes [41], a similar condition is fulfilled, i.e., **n** ≡ **N**, where **N** is the unit normal vector to the reference surface. This condition allows rewriting the relation (2) in the following form:(3)$$W_{b}=\int \frac{B}{2}{\left(C_{1}+C_{2}-C_{0}\right)}^{2}dS,$$

Where *C*_1_, *C*_2_ are the principal (geometric) curvatures of the reference surface; *C*_0_ is the spontaneous curvature of the lipid bilayer, which accounts for the possible compositional asymmetry of membrane monolayers. Here, it is taken into account that div(**n**) = div(**N**) = −(*C*_1_ + *C*_2_) = −*J*. This classical energy functional (3) has been effectively used to analyze a large number of membrane processes and structures [42,43,44]. It is often called the Helfrich functional [41].

The applicability of the Helfrich functional is very limited. The formal condition that the director coincides with the normal to the reference surface leads to the model in which the state of the membrane is completely determined by its shape. Thus, any internal structure of the membrane is excluded from consideration, and a membrane in the Helfrich model effectively becomes an infinitely thin structureless elastic film. The applicability of such a model is limited by small deformations only: both the difference between the geometric and spontaneous curvatures and the geometric curvature of the membrane in absolute value should be much smaller than the inverse thickness of the membrane: |*J* − *C*_0_|, |*J*| << 1/(2*h*) ≈ 0.25 nm^−1^, where *h* is the thickness of the lipid monolayer. This limitation is very significant since the characteristic curvature of membranes achieved in fusion processes often does not satisfy this criterion. When the curvature of the membrane is large, the state of its two monolayers will be different, and this difference cannot be taken into account in the framework of the model considering the membrane as an infinitely thin structureless film. Nevertheless, the attractiveness of the simplicity and effectiveness of the Helfrich functional led to a series of its successive modifications and generalizations. These modifications were not strictly justified. The most important of these generalizations was the application of the Helfrich functional not to the whole membrane, but separately to each of its monolayers [45]. In this case, the curvature is related to a certain surface passing inside the monolayer, the spontaneous curvature is also related to the monolayer, and the membrane deformation energy is represented by the sum of the deformation energies of its two constituent monolayers. Spontaneous curvature of lipid monolayer is the curvature of the monolayer surface, which it acquires in the absence of external forces and torques [46]. The concept of spontaneous curvature can be interpreted in terms of an averaged molecular shape of lipids. According to this interpretation, a positive spontaneous curvature corresponds to inverted conical lipids, such as lysolipids (large cross-sectional area of the polar part and small area of the hydrophobic part of the molecule), zero spontaneous curvature corresponds to cylindrical lipids, such as palmitoyloleophosphatidylcholine (POPC) (similar areas of hydrophobic and polar parts), and negative spontaneous curvature corresponds to conical lipids, such as dioleoylphosphatidylethanolamine (DOPE) (small cross-sectional area of the polar part and large area of hydrophobic part). Lipids with different spontaneous curvatures tend to form structures with different geometric curvature of the surface: lipids with positive spontaneous curvature form micelles, lipids with zero spontaneous curvature form flat bilayers, and lipids with negative spontaneous curvature form inverted lipid phases, for example, inverted hexagonal H_II_ phase [42,47].

To the best of our knowledge, the first work utilizing the generalized Helfrich model with an elastic energy functional written for separate membrane monolayers is devoted to a theoretical description of the membrane fusion process [48]. This model (also referred to as the Kozlov-Markin model) assumes, for the first time, that the fusion of bilayer membranes occurs sequentially, monolayer by monolayer. Previous hypotheses suggested that the membrane fusion could occur (i) by interdigitation of bilayers into each other; (ii) by breaking bilayers and reconnecting them in the new topology; (iii) through the formation of micelles in the contact region of bilayers, etc. [45,48,49]. However, these hypotheses were either formulated qualitatively, without assessing their energy feasibility, or were rejected as unrealistic just by the results of such an assessment [45,48,49]. In the Kozlov-Markin model, the membrane fusion is firstly considered as a multi-stage process, and a quantitative calculation of the energy of intermediate structures is provided. It is assumed that the first stage of this process is a merger of contact monolayers of two membranes resulting in the formation of a so-called stalk. The use of the Helfrich functional, Equation (3), for calculating the bending energy of contact and distal monolayers separately, made it possible, for the first time, to explain an experimentally observed dependence of the system evolution on the spontaneous curvature of monolayers of fusing membranes [42,50]. The generalization of the Helfrich model by applying it to individual monolayers of the membrane was so natural that it was not even explicitly formulated [48].

Since the work by W. Helfrich in 1973 [41], the theory of elasticity of lipid membranes was continuously developing and improving. The development of experimental techniques allowed observing new deformation modes, which were then introduced into the theories of elasticity. Approaches to the practical overcoming of natural limitations of linear theory associated with the requirement of small deformations of the membrane were developed [51,52,53]. Significant progress in the continuum theory was associated with the rapid parallel development of molecular dynamics methods, especially, the coarse-grained approximation [54,55,56]. These methods allowed both visualizing intermediate structures of the fusion process and determining the elastic parameters of lipid membranes [54,55,56,57,58]. These parameters can be used for calculations carried out in the framework of continuum theories. The consistent development of the theory of elasticity, together with an accumulation of experimental data on the structure and physicochemical properties of membranes, has led to the simultaneous development of new models of the fusion process, operating with new computational methods, and including more structural information about membranes. In this review, we focus on the parallel evolution of the theory of elasticity and models of the fusion processes in eukaryotic cells.

## 2. Results

### 2.1. Approaches to a Description of Membrane Fusion

Numerous excellent works have been published, detailing the classical concept of membrane fusion [27,46,59,60]. The progress in the accumulation and systematization of experimental data on the fusion process and the development of theoretical approaches describing deformations of lipid membranes resulted in the development of fusion models that did not fit into that classical paradigm. In the next section, we outline the main points and results of classical theories of membrane fusion. In the subsequent sections, we consider new fusion models designed to describe experimental data that could not be explained in the framework of classical theory.

A physical model of any biological process should not only explain the currently available experimental data, but should predict the results of future experiments as well, and allow one to plan experiments, which would determine necessary model parameters with the required accuracy. In the process of fusion, lipid bilayers of biological membranes inevitably severely deform, often with violation of their local integrity [23]. Their close approach, rupture, and further fusion do not proceed spontaneously, as the energy of thermal fluctuations is too small. For these processes to occur, some energy source, which is external to the membrane, is required. In biological systems, specialized proteins always participate in fusion processes [27]. The functioning of the influenza A virus fusion protein, hemagglutinin, during the viral infection seems to be the best studied. Another example is the protein complex SNARE, which facilitates the fusion of synaptic vesicles with the axon membrane during synaptic signal transduction [61]. Certainly, the vast variety of fusion proteins is not limited to these two examples; more fusion proteins of interest are described in the reviews [13,62,63]. At the same time, despite the large diversity of protein machines involved in fusion, it is common among all of them that their work covers energy costs necessary to remodel fusing membranes. Most biological membranes are alike: they possess similar elastic parameters, degree of hydration, and are surrounded by solutions of the same ionic strength, and exist in a relatively narrow pH range. Thus, one may expect that the process of membrane fusion and, consequently, the consumption of energy in its course should be similar in different cases of biological fusion. This allows developing rather general theoretical models focusing on membrane metamorphoses only and avoiding dipping deep into biochemical details of the conformational changes of fusion proteins. Briefly, in a general theoretical model, one constructs an energy functional where fusion proteins are not taken into account explicitly. The state of the membrane is determined by the physicochemical properties of lipid bilayers only. Further, by minimizing the energy one obtains a system of Euler-Lagrange differential equations. The influence of fusion proteins at various stages of the fusion process is described by the corresponding boundary conditions for these equations.

Physical models of membrane fusion can be separated into two kinds. In the first one, a separate intermediate structure (or several discrete structures) is hypothesized. By the structure, we mean a membrane state, at which its free energy has a local minimum. The energy depends on system parameters: membrane elastic properties, surface electric charge, solution ionic strength and pH, hydrophobic and hydrophilic interaction, etc. It is calculated relative to some global fixed reference state (usually it is planar membranes before the fusion). A common reference state allows comparing energies of different intermediate structures; however, the transitions between them cannot be described. Generally speaking, the structures can be separated by energy barriers of unknown arbitrary heights. Thus, we may make conclusions about the energetically preferable pathway for fusion but cannot say anything about its kinetic, about the rate of fusion. That kind of fusion description has been developed in numerous works [39,45,48,49,59,60]. Among them is the classical stalk fusion model [48,49], which is briefly summarized in the next section.

In the second kind of model, the fusion is treated as a continuous process of system remodeling from two initially separate intact membranes to a final single bilayer. In the framework of those models, a generalized single reaction coordinate is considered. Then at fixed reaction coordinate the energy of the membrane is optimized with respect to its degrees of freedom and the system’s energy along the reaction coordinate is thus calculated. As a result, the energy barriers separating intermediate structures and the total energy barrier of fusion can be estimated. The probability of the system transfer from the initial to the final state is proportional to the Boltzmann factor of the barrier height. A high degree of detailing makes the development of such models rather cumbersome [64]. In some works, short continuous trajectories between adjacent intermediate structures are modeled, rather than the trajectory of the entire fusion process. Models of this kind are designed in [64,65,66,67,68]. Those partial trajectories are built taking into account all known energy inputs. The total energy along the fusion pathway may be considered as an upper limit estimation. Indeed, despite a thoroughly developed model, we can still miss some degrees of freedom. An additional degree of freedom may only lower (may not increase, speaking strictly) a system’s energy. Herewith, there is always an unpleasant possibility that such a detailed description of the process is still insufficient, due to the incompleteness of available information on the physicochemical and elastic properties of the membranes. In that case, within a framework of a specific theoretical model, it is possible to take into account insignificant energy contribution and miss the major ones that are not yet available. As a result, the model that skips large unknown energy contributions may erroneously be accepted as energetically feasible, i.e., correct from a thermodynamic point of view.

Membrane energy during fusion is calculated using approaches that differ by the degree of modeling details. A significant part of the models uses a “macroscopic” continuum approach when the lipid membrane is considered as a continuous liquid crystal medium. In the most continuum models, to simplify calculations, highly symmetric intermediate states (usually axisymmetric) are considered [39,49,64,65,66,67,68]. Shape restrictions of membrane structures should generally lead to an overestimation of their calculated energies. Moreover, the characteristic spatial scale of the structures forming in the process of fusion is about a few nanometers, i.e., comparable to the lateral size of a single lipid molecule. Hence, a significant drawback of continuum approaches to the fusion description is their inevitable extrapolation to molecular spatial scales that is disputable for continuous models [69]. To make analytical computations in continuum models possible, linear elastic theories are usually used, which are applicable for small enough deformations only. However, calculated membrane deformations in the course of fusion can be so large that using the linear approximation to calculate membrane energy cannot be strictly justified. Surprisingly, it was recently shown that the linear generalized Helfrich model well describes strongly deformed membrane [53].

One should separately mention microscopic approaches based on the modern computational methods of numerical modeling, such as molecular dynamic of coarse-grain or all-atom lipid models or the Monte-Carlo method. Such approaches can hardly be called theoretical ones; rather, they should be attributed to “numerical experiments”. Numerical methods allow obtaining more detailed insight into structure and states of fusing membranes since they do not require “manual” building of intermediate states and enable visualization of these states directly during the simulation. However, a numerical experiment does not allow for analyzing the influence of general non-specific parameters of the system, such as the elastic properties of membranes or an external force applied by fusion proteins. These approaches deal with systems of a specific composition and under specific conditions. Additionally, due to computational power limitations, molecular dynamic methods make it possible to consider only simplified systems with certain artificial restrictions. Most molecular dynamic simulations are performed on a system of limited size; as a result, this approach makes it possible to study the fusion of very small vesicles with a diameter of 15–30 nm only [70], which approximately corresponds to the lower size limit of liposomes produced experimentally. Moreover, to speed up the fusion process in simulations sometimes artificial membrane modifications, like dehydration of lipid headgroups to force attraction between fusing membranes are applied [70]. This may lead to effective energy barrier reduction to ~10 *k_B_T* [71].

Another theoretical approach to calculate the fusion trajectory uses self-consistent field theory (SCFT) [72,73]. It is a microscopic approach that utilizes the statistical mechanics’ tools to obtain the membrane state and calculate its energy. Therefore, a molecule behavior is not simulated, but analytical calculations are performed as far as possible. In that model, both solvent and “lipids” are modeled by simple Gaussian chains consisting of two mutually repulsing kinds of “atoms”—hydrophilic and hydrophobic ones. “Lipids” are amphiphilic, i.e., they are formed from two kinds of monomers (diblock copolymers), and the solvent consists of hydrophilic segments only. No hydrophilic repulsion is implemented. SCFT approximation is applied to calculate the partition function, and then the free energy of the ensemble of Gaussian chains. Thus, it is quite an artificial model that probably can reveal qualitative rather than quantitative features of fusion [73]. Even under that approximation, the solution of final equations and, hence, the energy and the intermediate states can be found only numerically, and within some restrictions imposed on the system (e.g., axial symmetry).

In the next section, we review continuum models of fusion of different detailing, their evolution, modern state, and the results they lead to. Then, we describe the microscopic model approaches to fusion modeling.

### 2.2. Classical Fusion Model: Stalk, Hemifusion Diaphragm, Fusion Pore

Classical fusion model development started from the pioneering work of Kozlov and Markin [48]. In that study, the authors hypothesize that two bilayer membranes (Figure 1A) merge via successive fusion of monolayers: the contact ones first and then the distal ones. It is suggested that the monolayers brought to the close contact may locally rupture and expose membrane core to water (Figure 1B), creating opposing hydrophobic patches. The patches attract each other and finally merge resulting in an hourglass-like local connection between the contact monolayers—the so-called monolayer stalk [48,49]. The distal monolayers remain flat after this membrane remodeling. Historically the stalk was the first intermediate structure in fusion process modeling. In the subsequent works, radial expansion of the stalk is considered [68,74]. That process allows lipids of distal monolayers to come into tail-to-tail contact and to form the joint bilayer (Figure 1D). This structure is called hemifusion diaphragm or trilaminar structure. A rupture of the central bilayer of the diaphragm leads to the formation of the fusion pore, which connects two water volumes enclosed by the fusing membranes. The expansion of the fusion pore finalizes the membrane fusion process (Figure 1E) [74].

Before the development of the stalk model, the arsenal of the membrane elasticity theory was presented by the Helfrich model, which operated by a single deformation of bending—the only known membrane deformation at that time [41]. This model considers a membrane as a structureless infinitely thin film. However, it is clear that physical states of contact and distal monolayers of the stalk are different (Figure 1C). That was a reason to calculate the stalk energy in [48] in the framework of the generalized Helfrich model applied separately to the monolayers. In that model, a set of intermediate structures—stalk, hemifusion diaphragm, and fusion pore—was postulated. The shapes of deformed monolayers in the classical model have not been found through elastic energy optimization, but were postulated toroidal for curved parts of monolayers of the stalk, fusion pore, and fusion diaphragm. All intermediate structures in the stalk fusion model possess axial symmetry and mirror symmetry relative to an “equatorial” plane. The fusion process is represented by a sequence of intermediate structures, but transitions between them are not considered in detail. The state of each intermediate structure is determined by a single parameter, for example, the external radius of contact monolayers in the equatorial plane, *R*. The free energy as a function of *R* is calculated and analyzed. One important feature of that model is its “leakless”—during the fusion pathway “intracellular” solutions (up-most and down-most in Figure 1A) never continuously connect to the “extracellular” one. The leakless hypothesis allows the membrane to fulfill its barrier function during fusion, and it is supported by experiments on the fusion of two artificial bilayer lipid membranes (BLM). In that classical fusion experiment, two vertical parallel BLM are formed by the Mueller-Rudin method [75] on orifices in two Teflon partitions separating an experimental cell by three chambers (left, middle and right). The set of electrodes in all chambers allows the monitoring of membrane integrity (by capacitance control) and fusion event (by the increase of conductance between left and right chambers). Membranes are blown out toward each other by the application of hydrostatic pressure in the left and right chambers. In the middle chamber, the BLMs come into close contact, and after some delay, the trilaminar structure forms. It is registered both optically and by the electric capacity increase. The contact bilayer of the trilaminar structure is broken afterward spontaneously, or by the application of an electrical pulse. Final fusion is detected by an increase of the conductance between the left and right chambers. During these experiments, conductance between the middle and the outer chambers is never detected. In other words, the outer chambers are never connecting to the middle one [76,77].

Despite the simplicity of the stalk model, it has made it possible to explain all existing experimental data to date [42,50,78,79]. In particular, in the model experimental system, the merger of membranes is observed first, and only after a certain delay the water volumes unite, i.e., the fusion pore forms. Indeed, in the classical model, the stalk and hemifusion diaphragm are formed first, i.e., the membranes merge, and after that the fusion pore forms following the rupture of the central bilayer of the diaphragm.

The classical model predicts the dependence of the fusion rate on the membrane lipid composition or rather on the monolayer spontaneous curvature. It was detected in the model experimental system, that a positive spontaneous curvature, induced by exogenous addition of lysolipids into the middle chamber, slowed down the merger of membranes, while a negative spontaneous curvature promoted the formation of a stalk and trilaminar structure, that was in agreement with classical model predictions [80]. The positive spontaneous curvature of distal monolayers did not practically affect the hemifusion rate, but promoted the rupture of the trilaminar structure [80], again, in agreement with the classical model.

### 2.3. Evolution of Elastic Models

It later turned out that the classical fusion model is an example of a model that takes into account minor energy contributions, while major ones are ignored. The reason for that has been a lack of detailed information on the physicochemical properties of lipid membranes. That is why the model has been erroneously considered an energetically feasible one. In the postulated structure of the stalk the distal monolayers are planar [48]. This leads to the inevitable formation of cavities—voids—between contact and distal monolayers (Figure 1C). For the first time, D. Siegel [81] has pointed to these cavities. The energy of the void-lipid chain interface was calculated under the assumption that the void could be filled either with nothing (vacuum) or with water, or with air and water vapor. In all three cases, the void energy was several times higher than the elastic energy of membrane deformations for the classical stalk [81]. A similar circular void appeared at hemifusion diaphragm rim (Figure 1D), and, again, the void energy was greater than the energy of deformations. The additional energy was too high, and raised doubts about the feasibility of the whole model. D. Siegel’s works have led to the “energy crisis” of fusion theory: calculated energy was too high for membranes to fuse, but they effectively fused in experiments. On the other hand, the stalk model looked intuitively reasonable and agreed with experimental observations [82]. It was clear that to overcome the crisis, the voids should be reduced or eliminated. However, that was impossible in the framework of the Helfrich model with a single deformation mode of bending. To fill the void a new deformation mode—lipid tilt—was introduced [83]. This deformation is analogous to shear deformation, which is used in solid-body elasticity theory [84].

It has been suggested that the state of a lipid monolayer is determined not just by its surface shape, but by the orientation of lipid molecules as well. By analogy with the elasticity theory of liquid crystals, the average orientation of lipid molecules is characterized by a field of unit vectors **n** called directors [83]. This director field is defined at a surface, called the dividing surface, which is located inside the monolayer parallel to its outer surface. The shape of the dividing surface is defined by a field of unit normal vectors **N**. Bending deformation is defined by the geometrical curvature of the dividing surface. In the new approach, lipid molecule orientation is not restricted to be perpendicular to the surface, i.e., the director is not necessarily matched to the surface normal. In general cases, it can deviate from **N** by a certain vector **t** = **n/(nN)** − **N**, called the tilt vector. The elastic energy of the tilt deformation is proportional to the squared tilt vector, and the proportionality factor is called the tilt modulus [83]. The tilt deformation represents a new additional internal degree of freedom of a lipid monolayer.

The tilt deformation is not quite new, as it was already considered by W. Helfrich [41], who established that it was important on molecular (~1 nm) scale only, and could be neglected in all other cases. Later, the tilt was rigorously reintroduced to the membrane elasticity theory in the work [83], and virtually simultaneously in three studies [68,85,86] the tilt deformation has been used to solve the “energy crisis” of the fusion theory. In addition, the work [68] proposed a quantitative criterion for the possibility of spontaneous membrane processes. It has been shown that, for the membrane process to occur spontaneously, due to the energy of thermal fluctuations of lipids in a time of the order of several minutes, the height of the activation barrier of the process must not exceed approximately 40 *k_B_T* (*k_B_T* = 4.14 × 10^−21^ J).

The introduction of the tilt deformation allowed solving the energy crisis of the fusion theory. Due to this deformation, the void could be eliminated, and the calculated energy of the stalk turned out to be significantly lower than 40 *k_B_T*. Thus, it was shown that the stalk can be formed due to the energy of thermal fluctuations of lipids in a time of the order of several minutes, provided that the activation barrier of this process does not greatly exceed the energy of the final structure—stalk. Further calculations accounting for the tilt deformation have shown that the energy of the trilaminar structure is approximately 3.5 *k_B_T* per 1 nanometer of the perimeter of the central bilayer [46,74,87]. This means that even if the trilaminar structure is formed in the course of fusion, the radius of the central bilayer should not exceed about 1–2 nm [46,74,87]. According to the classical model [48,49], a through pore should form in the central bilayer to complete the fusion process. However, a detailed theoretical model of membrane poration that describes a continuous pathway from a planar continuous bilayer to a bilayer with a pore of a large radius has been developed much later [51,52,88,89]. In the classical model, the process of hemifusion diaphragm poration has not been considered in detail. Thus, even after the introduction of the tilt deformation and the solution of the “energy crisis” of the fusion theory, the classical model did not get rid of significant shortcomings. The most important of them was a consideration of a discrete set of intermediate structures, rather than a continuous sequence of intermediate states.

Formally, the Helfrich functional contains one more term associated with so-called saddle-splay or the Gaussian curvature [90,91]. This term takes into account the membrane elastic energy when the membrane is not flat but locally its total curvature is equal to zero. This happens when two principal curvatures have opposite signs and are equal by their absolute values at some point. The saddle splay energy is proportional to the product of the principal curvatures with a coefficient called the Gaussian modulus [92]. From the Gauss-Bonnet theorem [93], it follows that contribution of the Gaussian curvature to the total membrane energy depends only on its surface topology, and remains constant while the membrane conserves its connectivity. However, the fusion of two closed membranes leads to the formation of a single closed membrane. That means that the saddle splay energy halves too, and that must be taken into account to judge correctly on the energetic feasibility of a developing fusion model. In the classic fusion model, this contribution has not usually been taken into account as the numerical value of the Gaussian modulus was unknown. That modulus has been later measured in material and numerical experiments in [90,94,95,96]. As a result, the Gaussian curvature contribution turned to be substantial.

Contrary to bulk liquid crystals, lipid membranes are quasi two-dimensional films and can form closed vesicles [97]. The application of hydrostatic pressure to the interior of such a vesicle leads to membrane lateral stretching and the change of the area per lipid molecule [98,99]. The deformation of lateral stretching-compression is an additional degree of freedom. The elastic moduli: splay [98,99,100], lateral stretching [98,99], and tilt [100] are experimentally measured for membranes of different compositions.

In the linear theory of elasticity, small deformations are considered. The expression for the surface density of the elastic energy is obtained by expanding the free energy in a Taylor series up to the first non-vanishing (quadratic) order. If three small deformations are taken into account (splay, lateral stretching-compression, and tilt), then three cross-terms formally must enter into the elastic energy functional. Tilt deformation is characterized by a tilt vector, and vector quantities cannot linearly enter the expression for free energy. In this way, the cross-terms of tilt and splay deformations, as well as tilt and lateral stretching-compression, are eliminated. However, the cross-term over the deformations of splay and lateral stretching-compression should formally enter the energy functional with the corresponding elastic modulus. It was experimentally shown that a specific surface called neutral exists in a lipid monolayer; on this surface, the modulus of this cross-term vanishes [101], and thus, the splay and lateral stretching-compression deformations on this surface are energetically independent. This surface lies at the region of the junction of polar heads and hydrophobic chains of lipids, at a depth of approximately 0.7 nm from the outer polar boundary of the lipid monolayer [101]. Traditionally, in the theory of membrane elasticity, deformations and elastic moduli refer to the neutral surface. This allows one to calculate the deformation energies of various membrane structures using only three elastic moduli, corresponding to splay, lateral stretching-compression, and tilt.

The elastic energy functional with the additional term of lateral stretching-compression is successfully utilized to describe membrane deformation in a wide variety of studies, including studies of membrane fusion [64,66]. However, the completeness and correctness of that functional has recently been questioned and reconsidered in [102,103]. It was shown that it must be complemented by at least two contributions to elastic energy—twist and cross-term on curvature and tilt gradient. As far as we know, this stage of the evolution of the theory of elasticity of lipid membranes has not yet been reflected in the development of quantitative fusion models.

### 2.4. Hydration and Hydrophobic Interactions of Membranes

Lipid molecules are amphiphilic, and the outer surfaces of membranes are lined with polar groups of lipids. Polar groups orient water molecules in such a way that two polar surfaces in water repel at short distances. This phenomenon is called hydration repulsion [104,105,106]. When two flat membranes come together, the force of hydration repulsion between them increases exponentially [45,107]; this force referred to the unit area of the approaching surfaces is called disjoining pressure. The characteristic length of the hydration repulsion is *ξ_W_* ≈ 0.2–0.3 nm and depends on the particular lipid composition of the membranes [108]; the preexponential factor, according to various estimates, is equal to *P*_0_ ≈ 10^7^–10^10^ Pa [108,109]. Due to the large preexponential factor, hydration repulsion between the membranes is manifested at distances significantly exceeding the characteristic length *ξ_W_*. Previously, a universal theory was developed to calculate the energy of hydration interactions between polar surfaces [110]. This theory introduced a corresponding order parameter to parameterize water structure. The order parameter was considered equal to zero in bulk water and nonzero at the polar surface. Under the assumption of the smallness of the order parameter, the free energy was expanded in a Taylor series, following the Ginzburg-Landau formalism [110]. Unfortunately, the application of the developed theory to calculate the energy of hydration interactions between polar surfaces of arbitrary shape is difficult. An analytical solution of the resulting equations is possible only for systems with additional symmetry. In particular, analytical results were obtained for hydration repulsion of two flat surfaces [110], as well as for the energy of an infinitely long cylinder hollow filled with water [111]. 

Since the energy of hydration repulsion of two membranes is proportional to the area of their contact, it was already indicated in the early works that the close contact of fusing membranes is only possible locally [45]. It was assumed that symmetrical bulges with practically touching tips are formed on two opposing membranes. These bulges can both form as a result of the action of fusion proteins, and/or spontaneously, due to thermal fluctuations of the membrane shape [45].

It is assumed in [112] that the energy of hydration repulsion during the approach of fusing membranes can be minimized by the formation of point-like protrusion on one of the membranes, directed to the opposite membrane, which remains almost flat (Figure 2). Due to the shape of the protrusion, the distance between the membranes increases sharply, even at a small displacement from the protrusion tip. As a result, the hydration forces act only in the small region with the size of a single lipid molecule around the protrusion tip. Consideration of the point-like protrusion allows one to reduce the energy of hydration repulsion significantly, as compared to the case of two membrane bulges having a shape of a spherical segment with macroscopic radii [112]. However, the validity of the theoretical description of the point-like protrusion raises certain questions. The calculations in [112] are carried out in the approximation of a continuous elastic medium. However, the membrane structure at the top of the protrusion is assumed to be violated. This is modeled by a lower lateral density of lipid molecules, and the molecules near the protrusion tip are arranged discretely, almost manually. The formation of the point-like protrusion is impossible without applying to its top a point external force directed perpendicular to the membrane. However, during “normal” fusion, no possible source of such force can be found. The extraction of the point-like protrusion by fusion proteins between merging membranes is not possible for steric reasons. If the point of force application by fusion proteins does not coincide exactly with the protrusion tip, this application can only result in the formation of a smooth bulge with its tip having a shape of a spherical segment [113]. Smoothening of the protrusion tip shape leads to an increase in its area, which, in turn, causes a sharp increase in the energy of hydration repulsion between the membranes. Thus, the concept of the point-like protrusion cannot be physically realized, although it allows one to formally lower the energy of hydration repulsion between fusing membranes.

The local approach of the fusing membranes leads to a sharp increase in the energy density associated with hydration repulsion at the tips of the membrane bulges. It is assumed that this energy can be reduced by lateral displacement of the polar headgroups of lipids away from the area of tight contact of the membranes [45,114]. This results in the formation of small patches on the tips of the bulges, where hydrophobic lipid chains are exposed to water [115,116]. Such patches are called hydrophobic defects (Figure 2). Thus, due to lateral displacement of lipid polar groups, the energy of hydration repulsion decreases, but additional energy appears, which is associated with the exposure of hydrophobic chains of lipids and deformations of the lipid monolayer in the vicinity of the hydrophobic defect. The calculation of the deformation contribution to the energy is difficult, since, formally, in the framework of the accepted theory of elasticity of lipid membranes, deformations are related to the neutral surface of the lipid monolayer. However, this surface does not exist in the region of the hydrophobic defect. It is possible to work only in the region where the neutral surface does exist, but it is necessary to set certain values of deformations at the defect boundary. To choose a specific type of boundary conditions, one should use considerations external to the theory of elasticity, since this choice, in principle, cannot be justified within the framework of the utilized model.

The energy contribution associated with the exposure of hydrophobic chains of lipids to water is proportional to the area of the hydrophobic defect. The proportionality coefficient is the surface tension of the water-lipid chains interface, which is usually taken as being equal to the surface tension of the planar macroscopic water-decane interface, ~50 mN/m ≈ 12 *k_B_T*/nm^2^ [45]. However, it has been shown both experimentally [117] and theoretically [118] that the surface tension at the flat interface between a hydrophobic medium and water decreases sharply when approaching the identical interface. The energy of the system decreases as these surfaces approach each other. In other words, two parallel hydrophobic surfaces separated by a layer of water are attracted. The characteristic length of hydrophobic interactions is *ξ_h_* ~ 1 nm [45,117]. It is usually assumed that hydrophobic defects formed in two fusing membranes have the shape of flat coaxial circles of the same radius [65,66].

A consistent theoretical description of attracting circular hydrophobic defects formed at the tips of membrane bulges, which, in turn, repel due to hydration interactions, has not been yet developed. In the framework of the formalism constructed in Marčelja’s works [110,118] a change in the structure of water near hydrophobic and polar surfaces is described by a certain order parameter. However, there is no reason to assume that hydrophobic and hydration interactions should be described by the same order parameter, i.e., that the “hydrophobic” and “hydration” order parameters correspond to the same physical characteristic of water. Formally, it is necessary to consider two different order parameters—hydrophobic and hydration. To solve the Euler-Lagrange equations, allowing one to find the spatial distribution of the order parameters, it is necessary to set appropriate boundary conditions. Since the properties of macroscopic surfaces in water are well studied, boundary conditions for hydrophobic and hydration order parameters on hydrophobic and polar surfaces, respectively, are trivial. However, the determination of adequate boundary conditions for the hydrophobic order parameter on the polar surface, as well as for the hydration order parameter on the hydrophobic surface, causes serious difficulties.

At the circular boundary of the defect, an abrupt change in the surface properties of the membrane interacting with water formally occurs. In reality, the boundary of the defect can be significantly blurred due to the thermal motion of lipids. It is shown in [51], using molecular dynamics methods, that a hydrophobic defect may not have a clear boundary at all: the surface density of lipid polar heads is reduced in a certain region of the membrane, and the average density changes almost continuously from its value at an unperturbed membrane, to a value in the region of a hydrophobic defect. In this case, the defect boundary is introduced according to the Gibbs formalism: it is assumed that the polar groups of lipids are completely absent in the region of the defect, and the surrounding membrane is assumed to be unperturbed up to the boundary of the defect. The excess energy of the system obtained in such a model is attributed to the boundary of the defect. However, in the case of an abrupt change in the properties of the membrane surface, it is difficult to determine the value of this boundary energy. In the work [67], when describing a hydrophobic defect, it is assumed that, near the boundary of the defect, a transition zone exists of approximately one lipid wide, where neither hydration repulsion nor hydrophobic attraction act.

In addition, the displacement of polar heads of lipids from the area of tight contact leads to deformations of the surrounding membrane, which extend to several nanometers around the hydrophobic defect [66,113]. Moreover, the surface shape of the membrane near the hydrophobic defect can significantly deviate from the plane. This greatly complicates the calculation of the hydration repulsion of two membranes. To quantify the energy of the hydration interaction of surfaces of complex shape, in [119], the authors use the expressions obtained for the interaction of plane parallel polar surfaces; however, this potential is multiplied by the square of the cosine of the angle between the tangent planes to two bilayers. In general, a detailed and rigorous theoretical description of membrane interaction at short distances has not yet been developed.

### 2.5. Mechanisms of Fusion Protein Functioning

In real biological systems, fusion does not occur spontaneously; individual fusion proteins or their complexes always participate in this process [16,17]. It is believed that the main function of fusion proteins is to bring two lipid bilayers into a tight local contact, as well as to provide the energy required for driving the fusion process. The characteristic height of the overall fusion energy barriers can be estimated in a wide range: from several tens to hundreds of *k_B_T*, which cannot be overcome due to thermal fluctuations solely [68]. However, it was shown that if fusion proteins bring two bilayers into the close contact, the rest of the fusion process could proceed spontaneously at the expense of the energy of thermal fluctuations in a time of the order of several minutes, provided that the height of each activation barrier does not exceed ~40 *k_B_T* [68]. Thus, the energy stored in the fusion proteins covers the energy costs exceeding the “critical” 40 *k_B_T*. Therefore, for a detailed theoretical description of the fusion process, the energy contribution of the fusion proteins should be taken into account.

There is much evidence for the decisive role of hydration repulsion in the fusion process. Numerous experimental data manifest that the membrane fusion could occur spontaneously if substances bridging two merging membranes (e.g., Ca^2+^) or dehydrating their contact (e.g., polyethyleneglycol) are added between the fusing membranes [120,121]. These data make it possible to conclude that bringing two membranes into close contact and overcoming the hydration repulsion are the most energy-consuming phases of the fusion process, which cannot occur spontaneously at the expense of the energy of thermal fluctuations of lipids. As the fusion in such model systems further proceeds without participation of any proteins, one could conclude that the energy barriers on the rest of the fusion trajectory should not exceed the critical height of ~40 *k_B_T* [68]. Thus, the function of fusion proteins is rather analogous to the role of enzymes: they can lower the main activation energy barrier for fusion, and, probably, accelerate the forward rate constant of the process.

To bring membranes into close contact, fusion proteins must be well anchored to both interacting membranes. As a rule, a transmembrane domain (TMD) of a fusion protein is embedded into one of the fusing membranes. In synaptic fusion, one transmembrane part of the SNARE complex is anchored in the axon membrane (t-SNARE), while the second part of this complex, v-SNARE, is located in the synaptic vesicle membrane [122]. During fusion, these parts are combined into a trans-SNARE complex, converging neuronal and vesicular membranes. As a rule, the mechanical energy stored in one fusion protein molecule is insufficient to complete the fusion. For instance, synaptic fusion requires at least three SNARE complexes to cooperate their efforts [123]. In the case of the influenza virus, hemagglutinin forms trimers, and, according to various estimates, a fusion event requires the cooperative action of three to six trimers forming the so-called fusion rosette [38].

In the case of viral fusion proteins, in particular, the influenza A virus, the TMDs of hemagglutinins are located in the viral membrane [13]. As pH decreases, hemagglutinin undergoes a conformational transition, in which the N-terminal amphipathic fragment, called fusion peptide, leaves the hydrophobic pocket and attacks the target cell membrane, penetrating it to the depth of approximately one monolayer [124]. The conformational transition ends with the convergence of the C-terminal transmembrane domain and the N-terminal fusion peptide. If the fusion peptide incorporates into the target membrane, then, in the course of the conformational transition, the viral and cell membranes should come into close contact. Thus, in addition to inducing local deformation of the target membrane, fusion peptides act as anchors, through which fusion proteins exert mechanical forces on the target membrane [63,125]. If the distance between the fusing membranes is fixed far from the fusion site (for example, by “folded” hemagglutinins separating membranes by ~10 nm), then the local contact of membranes requires the formation of bulges facing each other. Already, in the first works on the classical model of fusion, it was shown that the energy of intermediate structures increases sharply with increasing intermembrane distance [48,68,85]. For hemagglutinin-mediated fusion, the formation of the bulges on the fusing membranes was observed experimentally [23,25]. A formation of a bulge in the target membrane can result from the membrane deformation by incorporated fusion peptides, and from pulling forces applied to the fusing membranes by fusion proteins [126,127]. It is believed that the first mechanism of bulge formation is mainly realized during a viral infection, in particular, in the hemagglutinin-mediated fusion of the influenza virus membrane with the endosomal membrane. The second mechanism occurs in the SNARE-mediated fusion of synaptic vesicles with the presynaptic membrane. However, in the case of influenza, the conformational rearrangement of hemagglutinin leads to tightening of the membranes [13], and, in the case of synaptic transmission, synaptotagmin, which is involved in the fusion, induces a negative spontaneous curvature in one of the membranes; such a spontaneous curvature is required for bulging [128,129]. Generally speaking, these mechanisms can be combined. It has also been hypothesized that the activity of fusion proteins is primarily aimed at the deformation of the target membrane, while the application of contracting forces to the fused membranes is secondary [62]. Such a view is oversimplified, and often turns out to be erroneous [130].

Numerous studies employing molecular dynamics methods have shown that amphipathic peptides incorporated into a lipid monolayer push apart polar heads of lipids [131,132,133,134]. This leads to the bending of the monolayer: from a flat one, it becomes locally convex. Thus, at the level of single molecules, fusion peptides reorient lipids near themselves, i.e., they create a jump (discontinuity) of the director at their boundaries. At the ensemble level, shallowly inserted amphipathic peptides induce a positive geometric curvature, i.e., they have a positive spontaneous curvature [135,136]. Accordingly, if there are a lot of fusion peptides in this system, together with the surrounding lipids, they can be considered as a continuous elastic medium, which differs from pure lipid monolayer by values of the elastic moduli and spontaneous curvature. Thus, the ensemble of fusion peptides can be considered as modifiers of the elastic energy functional of the lipid monolayer in a certain region of the membrane. In the case of a relatively small amount of fusion proteins in the fusion rosette, fusion peptides and membrane deformations produced by them should be considered explicitly. In this case, the elastic energy functional takes into account the characteristics of only the lipid monolayer, and fusion peptides determine boundary conditions for deformations. In terms of differential equations, when considered discretely, fusion proteins set boundary conditions for membranes at various stages of the fusion process. The energetics of the process is determined by the physicochemical properties of lipid bilayers, for which the energy functional is written and then the Euler-Lagrange differential equations are solved.

When fusion peptides are shallowly inserted into the contact monolayer of the target membrane, they induce a ring-shaped region of positive spontaneous curvature in the case of ring-like fusion rosette. In this case, it is impossible to form a bulge in the target membrane [113,129]. A bulge can only form if positive spontaneous curvature is induced in a circular, rather than annular region, or when negative spontaneous curvature is induced in the annular region [113,129]. Nevertheless, the hypothesis of the deformation of the target membrane by fusion peptides as a pre-requisite can be fruitfully used as a fundamental principle in constructing fusion models.

All known fusion proteins have transmembrane domains anchored in at least one of two fusing membranes. Any TMD has a distinct length, determined by the number of successive hydrophobic amino acids. This length may differ from the thickness of the hydrophobic core of the membrane, resulting in so-called hydrophobic mismatch [137]. If the membrane was flat up to the protein boundary, this would result in an exposure of either part of the hydrophobic TMD or the membrane hydrophobic core to the polar medium. As the contact of hydrophobic and polar media is energetically unfavorable, it is natural to assume that elastic membrane deformations should arise at the protein boundary, to compensate for the hydrophobic mismatch. Deformations require mechanical work to be performed; however, the corresponding energy is lower than the energy penalty of exposure of the hydrophobic medium to the polar one. The characteristic lateral length of deformations is about units of nanometers. When two protein molecules with TMDs possessing a hydrophobic mismatch with the membrane are far separated, the deformations induced by them are independent, and elastic energy is additive. However, if the proteins come closer, the deformations overlap, thus leading to an effective lateral interaction. The interaction is generally attractive, because if two proteins come into close contact, the total length of their boundary with the membrane should decrease, reducing the elastic energy. The dependence of the energy on the distance between two transmembrane proteins has been obtained in a large number of works [138,139,140]. In particular, the attractive energy profile is obtained both in MD simulations [141] and in the framework of continuum model [142] for a transmembrane dimer of gramicidin A. The hydrophobic mismatch is experimentally shown to drive the organization of syntaxin 1 and syntaxin 4 (participants of SNARE complex) into clusters in laterally homogeneous membranes [143]. The TMD of syntaxin 4 is somewhat longer than the TMD of syntaxin 1. Protein clustering is observed in membranes of different thicknesses, except one that fits perfectly the length of TMD. As TMD length is different for syntaxin 1 and syntaxin 4, these proteins form non-overlapping clusters at a given thickness of the membrane [143]. Such clustering could regulate the cooperativity of mechanical forces and torques induced by fusion proteins. In addition, it may induce local membrane bending, and potentially decrease the energy barrier for membrane fusion [143].

Cell membranes include hundreds of different types of lipids [144]. Under physiological conditions, lipids can undergo phase separation producing numerous microscopic domains with sizes varying in the range of 25–200 nm [145,146,147]. Similar domains in model membranes are more ordered than the surrounding membrane, and, consequently, they have higher elastic stiffness and bilayer thickness [98,148,149]. In addition, domains as small as tens nanometers in diameter are shown to be bilayer, i.e., to span the whole membrane [150]. It is higher elastic stiffness and larger bilayer thickness that are responsible for the bilayer structure of ordered domains [151,152]. The larger thickness of the domain results in the hydrophobic mismatch, which compensation is possible via elastic deformations arising at the domain boundary [152]. The energy of elastic deformations is believed to be a major contribution to the domain line tension, the interphase boundary energy related to the unit length of the boundary [153]. Ordered domains represent an inhomogeneity in membrane thickness, thus allowing transmembrane proteins to choose the membrane phase (ordered or disordered), where the bilayer thickness best fits the length of the protein TMD. Alternatively, transmembrane protein can induce local phase separation leading to the formation of a small lipid domain of optimal bilayer thickness in its vicinity by so-called wetting mechanism [154,155]. However, as the ordered domain possesses larger elastic moduli as compared to the surrounding membrane, the lateral distribution of transmembrane proteins is not controlled by the hydrophobic mismatch only. In the thorough analysis provided in the works [156,157], three key determinants of targeting of transmembrane proteins to ordered domains are found: (i) the length of TMD; (ii) the post-translational modification of the protein (palmitoylation, myristoylation); (iii) the surface area of the TMD side chains. The local enrichment of fusion proteins in a distinct membrane phase may regulate the cooperativity of their mechanical activity. In addition, it is demonstrated that most types of membrane inclusions, in particular, amphipathic and hydrophobic peptides, as well as transmembrane proteins with different TMD length and shape, manifest a strong affinity to the boundary of the ordered domain [157,158]., The optimal shape of the domain is a circular one, due to the line tension of its boundary. The affinity of TMDs and/or fusion peptides to the circular domain boundary may lead to the self-organization of fusion proteins into a ring-like fusion rosette, thus allowing concerting mechanical efforts developing by fusion protein molecules [65]. In addition, if the line tension of the domain boundary is high enough, it might cause bulging (and even pinch-off) of the domain out of the membrane plane [159,160], as this lead to a decrease of the length of the interphase boundary. Formation of bulges in merging membranes is an important stage of the fusion process; in a phase separated membrane the bulging may occur spontaneously, i.e., without consumption of the energy stored in fusion proteins.

### 2.6. Continuous Fusion Pathway: From Two Parallel Membranes to Stalk

The classical fusion theory is based on simple models of the lipid bilayer, which neglected the bilayer internal structure and postulated intermediate structure (first of all, their shape). Modeling of the fusion process in the framework of the continuum approach in its current state involves “guessing” of qualitative properties of intermediate structures, followed by optimization of their quantitative characteristics. In other words, this approach does not provide tools to find optimal intermediate structures; one can only suggest a set of possible system configurations and then estimate the physical (energetic) feasibility of the developed model. The monolayer stalk is a key universal fusion intermediate, confirmed experimentally [60]. That is why it is considered in the vast majority of theoretical models of fusion.

The initial state of the fusion process is two planar parallel membranes separated by a distance *D*; let us assume the membranes to be horizontal, for definiteness. To form the stalk, the membranes should be brought into close contact. The hydration repulsion obstructs this process; that is why it is energetically favorable to bring membranes together only locally, by forming two opposing bulges. In artificial systems, these protrusions may form spontaneously due to thermal fluctuations of the membrane shape [45]. In biological systems, specific fusion proteins provide bulging and approaching of the membranes [13,127,161]. Usually, a fusion model does not consider in detail a process of the bulge formation, but energetically justifies it. The energy density of hydration repulsion is maximal at the bulge tip. Lateral displacements of polar head-groups from this area yield small defects—hydrophobic patches at the tips of bulges [114]. This process may be energetically favorable when membranes are close enough: head-groups displacement replaces hydration repulsion by hydrophobic attraction. On the contrary, the lateral displacement of head-groups is unfavorable when the membranes are far separated, and the exposure of the monolayer hydrophobic core to water is energetically disadvantageous. The interplay of these energetic gains and losses forms a saddle-like energy surface. The defects usually are modeled by two identical coaxial disks [45,68]. The total system energy is a sum of three contributions: (i) hydrophobic attraction of the two defects in the opposing membranes; (ii) hydration repulsion of polar head-groups around the defects; (iii) membrane deformation, due to the reorientation of lipid molecules induced by head-groups displacements. The system state may be characterized by two geometrical parameters—the defect radius *r_h_* and defect separation *m*. The system energy *W*(*r_h_*, *m*) is minimal in the initial state when *r_h_* = 0 and *m* = *D*. Indeed, there is no hydrophobic core exposure into the water, no hydration repulsion when *D* is large enough, and no membrane deformation. Another energy minimum is possible when *m* = 0 and *r_h_* = *h*, where *h* is the thickness of the monolayer hydrophobic core. In that case, lipid molecules at the hydrophobic defect are horizontal and the defects themselves are connected (*m* = 0), i.e., hydrophobic lipid tails are not exposed into water. That structure is nothing else but the monolayer stalk—a highly curved structure. This means that the membrane separation increases quickly with distance from the stalk axis so hydration repulsion is small. On the other hand, the monolayer deformation energy is maximal. Hence, the energy minimum at *m* = 0, *r_h_* = *h* if exists is relatively shallow [66,68]. 

Moving membranes apart with open hydrophobic patches (*r_h_* = *h*, *m* → ∞), or bringing them together without opening the hydrophobic defects (*r_h_* = 0, *m* → 0) increases the system energy. Indeed, when *r_h_* and *m* are large enough the total energy is high as in that case the large hydrophobic patches are far away from each other and do not interact. Their energy is a result of pure exposure of lipid tails—hydrocarbon chains—into water. The surface tension of the water-tails interface is approximately equal to that at the water–decane interface ~12 *k_B_T*/mn^2^. For larger *r_h_* energy contribution of membrane deformation is considerable too. At small *r_h_* and *m*, the total energy grows quickly, due to the hydration repulsion of head-groups located in close proximity. Thus, the energy surface *W*(*r_h_*, *m*) has a saddle point at certain *r_h_^s^* and *m^s^*, defined from equations:(4)$$\frac{\partial W\left(r_{h}^{s},m^{s}\right)}{\partial r_{h}}=\frac{\partial W\left(r_{h}^{s},m^{s}\right)}{\partial m_{h}}=0.$$

The energy at saddle point *W*(*r_h_^s^*, *m^s^*) determines the height of an energy barrier at the optimal system trajectory from the initial state (*r_h_* = 0, *m* = *D*) to stalk (*r_h_* = *h*, *m* = 0). The entire optimal trajectory can be found for example by the method of gradient descent for *W*(*r_h_*, *m*) [59], when the optimal trajectory *m*(*r_h_*) satisfies the equation:(5)$$\frac{dm}{dr_{h}}=\frac{\partial W}{\partial m}/\frac{\partial W}{\partial r_{h}}$$
with the boundary condition *m*(*r_h_^s^*) = *m^s^*.

The optimal continuous trajectory of stalk formation is calculated in [63], taking into account bending deformation. Later, in [65], the trajectory is obtained with the account of splay and tilt deformations for membrane under lateral tension; and, in the most general case, with the account of saddle-splay and the area stretching/compression deformations in [66].

The resulting stalk radius is minimal and in the equatorial plane the radius is equal to a single lipid molecule length. For the fusion process to pass any further the stalk should expand [74,162]. In the pioneering study [48], the energy barrier of the stalk formation is approximately estimated as 100 *k_B_T*; only the deformation of contact monolayers was taken into account and their shapes are postulated to be toroidal. In [86], for the first time, the shape of the monolayer stalk is not fixed, but is calculated by minimizing the elastic energy that made it possible in certain cases to obtain even negative energy for the stalk, i.e., less than the energy of intact bilayers. However, in this work, the stalk is considered as an isolated structure, without calculating the trajectory of its formation from flat parallel membranes. The energy of the stalk of the minimum radius in [85] is estimated as ~45 *k_B_T*. In this case, the trajectory of the stalk formation was not calculated as well but an additional pre-stalk fusion intermediate is suggested—the point-like protrusion [112] (schematically shown in Figure 2). The estimation of the stalk formation energy barrier yielded ~50 *k_B_T*.

In [68], the continuous pathway of stalk formation is computed for the first time, taking into account lipid molecule splaying around the hydrophobic defects. Additionally, the action of fusion proteins is explicitly taken into account; it is assumed that fusion proteins provide the formation of two opposing semi-spherical bulges in fusing membranes. The energy barrier of stalk formation at the tips of these two bulges is 37 *k_B_T* [68]. A similar approach, using a more sophisticated theory of elasticity taking into account splay, tilt and saddle-splay [113], makes it possible to lower the energy barrier to 30 *k_B_T*. Explicit accounting of the work of fusion proteins led to the energy barrier of 20 *k_B_T*. The most advanced modern work [119] uses the so-called string model of lipids for calculations. This work can no longer be fully attributed to the continuum approach; in terms of the complexity of calculations and the details of modeling the system, it joins molecular dynamics calculations. In this work, for the height of the energy barrier of stalk formation the value of 25 *k_B_T* is obtained.

The described models make it possible to determine the dependence of the efficiency and rate of stalk formation on the lipid composition of fusing membranes. The lipid composition determines the physical characteristics of the membranes: elastic moduli, thickness, spontaneous curvature of the monolayers, etc. The majority of theoretical models predict that both the stalk energy and the height of the energy barrier of its formation should decrease with the negative spontaneous curvature of contact monolayers of merging membranes, and increase with their positive spontaneous curvature [42,85]. This prediction is supported by experiments on the fusion of flat model bilayers. It was found that lipids with negative spontaneous curvature, such as DOPE, accelerate the formation of the stalk; lipids with positive spontaneous curvature, for example, lysolipids, on the contrary, slow it down [78]. A similar dependence of the hemifusion rate on the lipid composition is also detected in numerous experiments in various systems ranging from Ca^2+^-induced fusion to the cell infection with enveloped viruses [80,163]. Such an agreement of theoretical and experimental results, even at a qualitative level, is a strong argument in favor of the stalk as an intermediate structure of the fusion process. Unfortunately, it is much more difficult to achieve a quantitative agreement, because varying the lipid composition modulates not only the spontaneous curvature of the membrane and its elastic moduli, but some other membrane parameters as well. For example, it has been experimentally shown that membranes formed from various lipids have different hydration repulsion parameters (characteristic length and disjoining pressure) [108]. The matter is also complicated by the fact that, in most cases, a physiologically relevant change of system parameters leads to a relatively small change in the height of the energy barrier, by a value comparable to the influence of the experimental error of the elastic moduli determination [98,99]. For example, according to [119] a variation in spontaneous curvature by 0.05 nm^−1^ that corresponds to a 25% change of cholesterol content in the membrane leads to a change in the energy barrier height by 5 *k_B_T* only. At the same time, variations by several *k_B_T* of the Boltzmann factor lead to an order of magnitude change of the measured characteristics, for example, the probability or waiting time for stalk formation. Such a scale of the height of the energy barrier variation—units of *k_B_T*—is representative of the investigated process in general. For example, it is shown in [66] that a change of pH of aqueous membrane environment leads to a change in the height of the energy barrier of stalk formation by 4 *k_B_T* that, however, affects significantly the average hemifusion waiting time measured experimentally.

Protein action significantly influences the energy barrier of stalk formation. Fusion protein can pull the opposing membranes together. A more sophisticated action is membrane modification by peptides in the vicinity of the fusion site. In the framework of the continuum approach, this is modeled by a director discontinuity at the peptide or protein edge, or by induction of spontaneous curvature [126]. In a series of studies [60,130] the last mechanism is declared as a universal one. Shallow insertion of the amphipathic peptide into the target membrane has an effect resembling the induction of positive spontaneous curvature. However, the positive spontaneous curvature induced by lipids (e.g., lysolipids) promotes pore formation [80,164] and inhibits stalk formation and fusion. Hence, the action of fusion proteins with amphipathic fusion peptides cannot be based entirely on the modification of a contact monolayer of target membrane, and, probably inevitably, requires an application of forces pulling the membranes together. To decrease the energy barrier of stalk formation fusion peptides should efficiently induce the negative spontaneous curvature [80], for which their insertion depth should be increased. The effect of insertion depth is qualitatively illustrated in Figure 3: in the ring area of the fusion rosette, a shallow insertion induces positive spontaneous curvature locally, and facilitates the formation of a through pore; a deep insertion induces negative spontaneous curvature and creates a bulge.

Calculations explicitly taking into account the depth of fusion peptide incorporation are performed in [67,113]. They confirm these conclusions. Moreover, experimental data for HIV fusion peptides indicate that the deeper the insertion into the target contact monolayer, the more probable the fusion is [165]. One should mention that the real penetration depth of the fusion peptide of influenza virus hemagglutinin is still disputable. The majority of papers report that these peptides are amphipathic; they penetrate rather shallowly and hence induce a positive spontaneous curvature [166]. However, there are contradicting data, reporting that fusion peptides penetrate deeply and induce negative spontaneous curvature [167]. The work [113] theoretically demonstrates that direct pulling of membranes together by fusion proteins plays a leading role in a lowering of the height of the energy barrier of stalk formation. It is found that it is impossible to significantly reduce the barrier height regardless of the sign and magnitude of the spontaneous curvature induced by fusion peptides without the application of a pulling force. The pulling force is generated during the conformational transition (coiled-coil formation) of fusion proteins when the conformational energy stored in the proteins is released as mechanical action. For the influenza virus hemagglutinin, this energy is estimated as 30 *k_B_T* per trimer [168].

### 2.7. The Continuous Trajectory of the Fusion: From Stalk to Fusion Pore

At the stalk stage, two contact monolayers of merging membranes fuse, while distal monolayers are isolated from each other (Figure 4B). Near the equatorial plane of the stalk, only lipids of fused contact monolayers are oriented horizontally, while the lipids of distal monolayers are oriented approximately vertically (Figure 4B). In the final structure of the fusion process, which is the fusion pore, in the vicinity of the equatorial plane lipid molecules of both contact and distal monolayers are oriented horizontally (Figure 4D,E). The assumption of leakless fusion process requires introducing several intermediate states, which allow continuous switching from the vertical orientation of lipid molecules of distal monolayers of the stalk to the horizontal orientation at the fusion pore equator in the final stage. Since the lateral size of lipid molecules is much smaller than their length, for the continuous reorientation of lipids of distal monolayers, it is necessary to “spread” fused contact monolayers in the radial direction by the distance not less than the length of a lipid molecule. In this scenario, the reorientation of lipids of distal monolayers leads to the formation of the fusion pore of zero lumen radius. In the classical model of fusion, an intermediate structure that allows for the continuous reorientation of lipids of distal monolayers is hemifusion diaphragm or trilaminar structure (Figure 4C). In this structure, due to the radial dislocation of merged contact monolayers, lipid molecules of distal monolayers come into tail-to-tail contact, forming a unified single central bilayer. It is assumed that a through hydrophilic pore may spontaneously form in the central bilayer (Figure 4D).

In reference [74], a continuous trajectory of the transition from the stalk to the hemifusion diaphragm is obtained, taking into account splay and tilt deformations. It is shown that spontaneous expansion of the diaphragm is possible only in the case of large negative spontaneous curvature of monolayers of the fusing membranes. The required spontaneous curvature is equal to −0.3 nm^−1^, which is close to the spontaneous curvature of pure DOPE or cholesterol [169,170]. In a more realistic case of membranes formed from dioleiolphosphatidylcholine (DOPC), which have a spontaneous curvature of about −0.1 nm^−1^ [169,170], the energy of the hemifusion diaphragm almost linearly depends on the perimeter of its central bilayer, with a proportionality coefficient of approximately 3.5 *k_B_T*/nm. In such a membrane, the energy of the smallest stalk is approximately 40 *k_B_T*. If the radius of the central bilayer of the hemifusion diaphragm is equal to the length of a lipid molecule (1.5–2 nm), its energy reaches ~80 *k_B_T*. According to theoretical estimates [68], the energy difference (80 − 40 = 40 *k_B_T*) can be covered by thermal fluctuations of lipids. However, at the radius of the central bilayer of ~1.5–2 nm, the energy of the hemifusion diaphragm does not have a local minimum [74]. This means that thermal fluctuations have to cover the energy cost of both the expansion of the diaphragm and the formation of the through pore in the central bilayer (Figure 4D). It is also shown in the work [74] that membrane deformations at the hemifusion diaphragm rim induce significant lateral tension in the central bilayer, reaching about 12 mN/m. It is suggested that such lateral tension leads to the rupture of the central bilayer, and the formation of the through pore. A model that makes it possible to calculate continuous trajectories of through pore formation in membranes is developed in [51,52]. According to this model, the formation of the through pore in the membrane made from DOPC requires overcoming the energy barrier of a height of approximately 35–40 *k_B_T*. This barrier is mainly determined by the surface tension of the water-hydrophobic core interface, and it is practically independent of lateral tension. The energy barrier of pore formation is lower for lipids with a more positive spontaneous curvature; however, in this case, the energy cost of radial expansion of the hemifusion diaphragm increases. With significant negative spontaneous curvature of merged membranes, the radial expansion of the diaphragm occurs spontaneously; however, in this case, the energy barrier of pore formation in the central bilayer sharply increases. Thus, the classical fusion model, which considers the hemifusion diaphragm as an intermediate structure, is not energetically feasible. Its main problem is conditioned by that the radial expansion of the hemifusion diaphragm and formation of a through pore in the central bilayer are considered as two successive processes, and these processes contribute additively to the height of the effective energy barrier of fusion pore formation.

The high energy cost of radial expansion of the hemifusion diaphragm inspired the development of an alternative model that considers direct transition of the stalk to the fusion pore, bypassing the stage of hemifusion diaphragm [68]. Under the assumption of reversibility of fusion, the reverse process was analyzed, i.e., the transition of the fusion pore into the stalk. The outer radius of contact monolayers in the equatorial plane, *R*, was used as the process coordinate. With decreasing *R*, the radius *R_L_* of the fusion pore lumen also decreased. It was assumed that, upon reaching *R_L_* ~ 0.5 nm, powerful hydration repulsion forces begin to act inside the pore lumen, preventing a further decrease in *R_L_*. If *R* continues decreasing, the bilayer forming the sidewall of the fusion pore is forced to compress. Such a structure is called a compressed fusion pore. It was shown that, at a certain radius *R*, the stalk and the compressed fusion pore have equal energies, and a transition between them is possible at this point [68]. The height of the energy barrier of the transition from the stalk into a compressed pore is approximately 40 *k_B_T*. However, it was assumed that the transition occurs abruptly, and the continuous transition trajectory was not calculated. This means that the energy barrier should be complemented by some energy required for reorienting and compressing the lipids of the distal monolayers, which inevitably makes this model of fusion energetically unfeasible. In addition, the model of a compressed fusion pore predicts the dependence of the process rate on the spontaneous curvature of monolayers of fusing membranes [171], which is the direct opposite of the experimental observations [42,50,80].

To unambiguously judge the energy feasibility of a model of the fusion, it is necessary to obtain the full energy profile along a continuous reversible trajectory of the process. Such a trajectory is constructed in [64]. It is assumed that a cylindrical hydrophobic defect can form along the axis of rotational symmetry of the stalk. This defect is considered as the initial stage of the formation of a through pore [51,52]. Hydration interactions are explicitly taken into account near the boundary of the hydrophobic defect. Thus, three contributions into the energy of the system are considered: membrane deformations (splay and lateral stretching-compression), hydrophobic, and hydration interactions. The state of the system was characterized by two parameters: the hydrophobic defect radius *r_h_* and the external radius of contact monolayers in the equatorial plane, *R*. It was assumed that the height of the hydrophobic cylinder *L* is uniquely determined by the values of *r_h_* and *R*. During the expansion of the stalk *L* monotonously decreased, reflecting a gradual change in the orientation of lipids of distal monolayers from vertical in the stalk state to horizontal in the fusion pore state. Thus, the model [64] considers a simultaneous expansion of the stalk and formation of the through pore. The energy surface *W*(*r_h_*, *R*) has a saddle point, which is found by an analog of Equation (4). The equation of the optimal trajectory *r_h_*(*R*) is found similarly to Equation (5). The optimal trajectory passes through the saddle point of the energy surface, which determines the height of the energy barrier of the stalk to the fusion pore transition. It was found that the energy barrier height for merging membranes with zero spontaneous curvature of monolayers is approximately 40 *k_B_T*. The height of the energy barrier should increase if the spontaneous curvature of distal monolayers is negative, and should decrease if it is positive, in good agreement with the available experimental data [42,50,80]. In the considered work, the shape of the monolayer surfaces was not optimized and was considered as toroidal, i.e., the obtained energy value is an upper estimate, since the optimization of the membrane shape can only lower it. Note that the fully optimized shape of the fusion pore was calculated in [172]. It was found that bilayers in the fusion pore have a shape of teardrops, i.e., the distance between membranes slowly increases while approaching the system’s rotational axis, and then it decreases sharply, i.e., the intermembrane distance depends on the radial coordinate non-monotonically. The model developed in the work [64] for the transition of the stalk to the fusion pore allows finding a continuous trajectory of the process. The calculation of the energy of the system along the trajectory confirmed the energy feasibility of this model. Its main feature is that, in contrast to the models described in [68,74], the radial expansion of the stalk and the formation of the through pore in the central bilayer formed by distal monolayers of fused membranes do not occur sequentially, but simultaneously; this leads to a significant decrease in the height of the transition energy barrier.

In summary, the improvement and development of the classical model of the fusion process by detailing contributions to the energy of the system and the evolution of the theory of elasticity of lipid membranes made it possible to construct continuous trajectories of the process from the initial state of flat parallel membranes (Figure 4A) to the final state of the fusion pore (Figure 4E). The calculated heights of energy barriers of trajectories indicate the energy feasibility of developed models [68,74], which makes it possible to declare the final solution of the “energy crisis” of fusion theory [85]. Predicted dependences of heights of energy barriers on elastic parameters of fusing membranes are in a good agreement with available experimental data [42,50,80].

The described models were developed under the assumption that membrane fusion occurs without leakage, which is probably true for many fusion events. Thus, fusion in synaptic transmission seems to be completely leakless [28,29]. However, there is considerable experimental evidence that at least in the case of fusion induced by enveloped viruses, both viral and target membranes can rupture, i.e., fusion in such systems may be leaky [173,174,175,176]. The continuum approach used in the developed models allows one quite effectively to describe intermediate structures: stalk, hemifusion diaphragm, and fusion pore. However, the description of the continuous transition between these structures within the framework of this approach raises conceptual problems. In these models, linear continuum theories are applied at subnanometer scales to calculate the energy of strong deformations, i.e., beyond the formal limits of applicability of these theories. The applicability of the linear continuum approach under such conditions is verified experimentally, as well as by the methods of molecular dynamics [53,58]. It is shown that the Helfrich model allows one to quantitatively describe strongly deformed membrane structures, but only when applied monolayerwise [53]. In addition, fundamental problems arise in the modeling of hydrophobic defects within the framework of continuum theories. Simplified modeling of key intermediates leads to artificial restriction of the considered trajectories of the evolution of the system, and results in incorrect and incomplete theoretical estimates. To describe and predict other hypothetical intermediates that do not fit into the classical paradigm, it is necessary to study the fusion process at a level much more detailed than the continuum approach provides. Visualization and high (atomic scale) detailing of the structure of intermediate states can be achieved by the methods of molecular dynamics [162,177,178]. The results obtained by these methods can be used to develop more general continuum models.

### 2.8. Molecular Modeling of the Fusion Process. Non-Classical Intermediates

The process of membrane fusion has been actively studied by molecular dynamics methods both by coarse-grained [54,55,162] and all-atomic approximations [179,180], as well as using the Monte Carlo simulations [181]. In these works, molecular structural details of intermediate fusion structures are elucidated, and energy barriers of their formation are evaluated. Although the use of continuum models does not require assumptions of the radial symmetry of considered membrane structures, practical calculations of structures that do not have any symmetry are fraught with considerable difficulties. At the same time, molecular modeling methods are not limited to symmetrical membrane structures. The only significant restriction on the symmetry of the simulated structures is imposed by periodic boundary conditions, which are used, as a rule, in molecular dynamics. Periodic boundary conditions require mutual compensation of all bending torques applied to the membrane inside one cubic simulation cell. The influence of periodic boundary conditions can be reduced by modeling large systems, but this, in turn, increases the complexity of modeling.

Molecular dynamics make it possible to observe not only metastable structures, but also the transitions between them. In several studies, the formation of the stalk from two flat parallel or locally spherical membranes has been simulated. When membranes approach each other, their local flattening has been observed [177]. Then hydrophobic defects can form in contact monolayers; the area of the defects is comparable to the lateral area of several lipid molecules [55], which is consistent with calculations of hydrophobic defect structure carried out in the framework of the continuum approach [66,68]. In some studies, a single lipid molecule two hydrophobic chains of which are incorporated into two opposed bilayers—each chain into its own bilayer—provides the primary contact of membranes [180]. In this state of two membranes connected by a single lipid molecule, the free energy of the system has a local minimum [56], i.e., this state is a metastable intermediate structure. The results of this work inspire the development of alternative detailed trajectories of stalk formation in the framework of the continuum approach. The energy barriers on the alternative stalk formation trajectory obtained by MD methods are comparable with the barriers calculated from continuum models and equal to ~20 *k_B_T* [56].

The evolution of the stalk to fusion pore through a structure similar to the small hemifusion diaphragm is observed by molecular modeling methods in the works [71,182]. In addition, alternative trajectories of the fusion process are registered. In several studies, a temporary linear, rather than radial, expansion of the stalk is observed [55,162,181,183]. However, the final hemifusion structure possesses a rotational symmetry, similar to structures formed during the radial expansion of the stalk in the classical fusion model. Sometimes, during the linear expansion of the stalk a short-lived through pore forms in one of the membranes in the region of the linear stalk (Figure 5B) [55,181,183]. In references [55,183], the through pore quickly closes, resulting in a small amount of water encapsulated inside a small vesicle formed when a linear stalk collapsed into a circular-like stalk. The resulting structure looks like two initial flat membranes forming two hemifusion diaphragms near the upper and lower poles of the small central vesicle [183]. The formation of such structure is accompanied by the poration of one of the membranes, i.e., fusion along this trajectory is leaky. Similar structures are observed in several studies using molecular dynamics methods (see reviews [70,184]). The membrane of the central vesicle is subjected to strong elastic stress; this leads to a high probability of its poration. An asymmetric hemifusion diaphragm forms as a result of the formation of the through pore at the upper or lower poles of the central vesicle (Figure 5C). After that, the fusion process moves almost along the classical trajectory [162,181]. In the alternative trajectory, not one, but two pores are formed near the linear stalk, one in each of the fusing membranes [70]. As a result of enveloping two through pores by a linearly expanding stalk, a fusion pore immediately forms, bypassing the hemifusion diaphragm stage [55].

The specific trajectory of the fusion process is determined by heights of energy barriers on various possible trajectories. In the classical scheme, the stalk expands radially to a small hemifusion diaphragm followed by a through pore formation in its central bilayer. The energy barrier of this trajectory is determined by the energies of the hemifusion diaphragm and the pore. According to the alternative path, the linear expansion of the stalk takes place, and then it is followed by the formation of a pore in one or both fusing membranes. The energy barrier of the alternative trajectory is determined by the energy of linear expansion of the stalk and the energy barrier of the pore appearance. The values of energy barriers, and therefore the preferences of a particular fusion path, are determined by elastic properties of the membranes: the elastic moduli, the spontaneous curvature of lipid monolayers, etc.

In [162], utilizing molecular dynamics methods for the POPC membrane (approximately zero spontaneous curvature), it is shown that the energy barrier on the alternative trajectory is ~35 *k_B_T*, while the classical trajectory has a barrier ~17 *k_B_T*. When 40% of palmitoyloleoylposphatidylethanolamine (POPE) with a significant negative spontaneous curvature is added to the membranes, the fusion energy barrier along the alternative path decreases to 15 *k_B_T*, while the energy barrier on the classical path slightly increases to 18 *k_B_T*. As a result, the alternative trajectory becomes more energetically preferable. At the same time, the addition of 30% of cholesterol, which increases the bending stiffness of membranes and has a significant negative spontaneous curvature [169,170], leads to an increase of the energy barrier on the alternative trajectory to 64 *k_B_T*, and to 25 *k_B_T* on the classical trajectory.

### 2.9. Leaky Fusion: Experimental Observations and Continuum Modeling

Alternative trajectories of the fusion process associated with the formation of leaky intermediates were systematically studied by molecular dynamics methods. It was numerical experiments that made it possible for the first time to visualize leaky intermediates and determine their molecular structure [185]. At the same time, experimental data were accumulated on the occurrence of leakage in the course of fusion processes in a wide variety of systems [175,176,186,187]. To the best of our knowledge, the leakage was never observed in the fusion of neurotransmitter-loaded vesicles with the presynaptic membrane. Interestingly, experimental data indicates that leakage can occur at different stages of the fusion process, both after the appearance of the stalk and prior to it. The stalk formation in such experiments is usually detected using fluorescently labeled lipid analogs. During the stalk formation, contact monolayers of membranes unite, and a fluorescent dye can flow from one membrane to another. Typically, the dye is loaded into one of the merging membranes at a concentration of self-quenching. When the stalk is formed, the dye is diluted, due to diffusion into another membrane, which is registered as an increase of fluorescence intensity [188]. The formation of through pores in fused membranes, as a rule, is detected using aqueous fluorescent dyes or by electrophysiological methods [175,187,189]. As far as we know, in all experiments where the leakage is observed, the process of membrane fusion is induced by fusion proteins [173,175,190,191]. In the course of fusion of model membranes without the participation of fusion proteins, the leakage is not practically registered. This makes it possible to suggest that through pores in fusing membranes are induced by fusion proteins.

In [187], fluorescence microscopy reveals leakage during the fusion of giant unilamellar vesicles with influenza virions. The leakage occurs at the stage of the hemifusion diaphragm formation. The dependence of efficiency of membrane poration on the spontaneous curvature of monolayers is experimentally established. An increase in spontaneous curvature leads to an exponential increase in the number of formed through pores. The leakage is also observed in the course of the fusion of red blood cells with cells expressing hemagglutinin of the influenza virus [175]. Formation of through pores is registered by means of electrophysiology. It is shown that in this system the leakage occurs transiently at the stage of formation of a small fusion pore in 70–90% of experiments, but usually the through pores quickly reseal. Leaky structures formed after the stalk stage are also observed by cryoelectron microscopy in the work [23]. It is demonstrated that the probability of the formation of such leaky structures is determined by the lipid composition of the target membrane.

Some experimental data indicate the possibility of poration of fusing membranes before stalk formation. For example, in the works [173,174] the leakage of aqueous fluorescent dyes from vesicles is observed during the fusion. The kinetics of the leakage differs very little from the kinetics of the fusion. This suggests that leakage occurs in the early stages of fusion, and not during or after the formation of the hemifusion diaphragm. Membrane lysis is detected in the case of yeast cell fusion and SNARE-dependent fusion of vacuoles [176,192]. In the work [192], the through pore formation is associated with the excessive force exerted on membranes by overexpressed fusion proteins. Formally, this can be interpreted as the formation of a leaky intermediate at the very initial stage of the fusion [193].

In the case of the influenza virus, the insertion of fusion peptides into the target membrane plays a decisive role in its possible poration. In the work [194], a leakage in the target membrane is registered even before the merger of contact monolayers of the membranes and stalk formation. The necessary condition for these observations is a mutation of the fusion peptide, leading to its shallower insertion into the target membrane as compared to the wild-type. More shallow insertion of peptides should lead to an increase of induced spontaneous curvature of the contact monolayer. In the work [195], authors perform molecular modeling of the interaction of such mutant peptide with the membrane. They show that the effect of the mutant peptide on the membrane is similar to that of amphipathic antimicrobial peptides. Under certain conditions, such peptides induce the formation of through pores in the membranes [196,197,198]. Membrane structures that can be considered as pre-stalk leaky intermediates are visualized by cryoelectron microscopy in the work [199]. The membrane appeared to be strongly deformed in these structures. It is assumed that severe deformation of the target membrane could cause the formation of through pores in it (see Figure 6B). Similar pre-stalk leaky intermediates are observed in several experimental studies [23,200]. It follows, from cryo-EM images, that such intermediate structures possess axial symmetry: an approximately circular pore in the target membrane is surrounded by a ring of fusion peptides, while transmembrane domains of fusion proteins form a ring-like structure in the viral membrane (Figure 6B). This structure was called a π-shaped structure [23]. The presence of axial symmetry makes it possible to describe this configuration in the framework of the continuum approach.

Thus, the available experimental and molecular dynamic data suggest a robust alternative to classical trajectories of the fusion process, formulated as a new fusion paradigm [201]. In some studies, the leakage during stalk formation is even declared as an essential feature of the fusion process [186]. In the theoretical work [202], in the framework of the continuum self-consistent field theory, it is shown that in some cases the energy barrier of the fusion process on the leaky trajectory is several *k_B_T* less than on the classical path.

The possible presence of through pores in fusing membranes greatly complicates the description of intermediate states and structures of the fusion process. To determine the most probable trajectory of the process, it is necessary to calculate and compare the energy barriers on the trajectories. The solution to this problem is possible only with the correct modeling of hydrophobic defects in membranes. For this, a detailed theoretical description of the formation of through pores in flat membranes is developed and continuous trajectories are obtained, leading from an intact bilayer via a hydrophobic defect to a through hydrophilic pore [51,52]. It is shown that the shallow insertion of fusion peptides should significantly facilitate the formation of through pores [203].

In the work [67], a novel theoretical model for the fusion of membranes induced by the influenza virus is developed. The model takes into account the possibility of the formation of both the stalk and the leaky pre-stalk π-shaped structure. The authors show that the depth of insertion of fusion peptides into the target membrane generally determines the probability of formation of the π-shaped structure. A decrease in the insertion depth leads to an increase in the energy barrier of the stalk formation and, at the same time, to a decrease in the energy barrier of the formation of through pores in the target membrane. In the case of a shallow insertion of fusion peptides, the poration of the target membrane becomes more feasible than the fusion. The results obtained in the work [67] explain experimental data that indicate a correlation between the depth of HIV fusion peptide insertion and its fusion efficiency: the deeper the insertion, the more efficient is the membrane fusion [165]. In addition, the work [67] qualitatively explains the results of the paper [194], in which the leakage is observed only in the case of a fusion peptide mutation, resulting in a decrease of its insertion depth into the target membrane.

## 3. Conclusions

The pioneering theoretical models of the fusion of biological membranes were developed about 40 years ago. Starting with the simplest theories that consider lipid monolayers as structureless surfaces characterized by bending stiffness only, fusion models became continuously more complicated as new data on the structure and physicochemical properties of membranes accumulated, the theory of elasticity of lipid bilayers improved, and a theoretical description of hydrophobic and hydration interactions was developed. The development and improvement of continuum models have led to the possibility of describing strong deformations arising at molecular scales [53]. These developments made significant progress in theoretical models of membrane fusion possible. New results obtained in recent in vivo, in vitro, and in silica studies could not be described in the framework of the classical model of fusion pathway [23,187,202]. These data show that strongly asymmetric membrane structures can form during fusion, and the process may be accompanied by loss of the membrane barrier function, i.e., it may occur with a leakage. The leakage could be transient, with the fusion process eventually ending with the formation of a fusion pore. Alternatively, a long-lived through pore can form in a membrane. This pore can be stabilized by fusion peptides, and the fusion process will stop at the formation of the π-shaped structure. Initially, the formations of leaky intermediates during the fusion were treated as minor and artificial events, which were not accepted for consideration, in part due to the scarcity and complexity of their modeling. However, with the development of experimental and computational methods, these structures were visualized and studied in molecular details. Leakage intermediates, as a rule, do not have axial symmetry, but a combination of detailed continuum models and numerical methods allowed describing such structures [67,202].

One of the challenging problems of the continuum approach to the fusion process is the development of a theoretical description of the very initial stage of the interaction of lipid bilayers through a thin water layer, which is still modeled in the framework of the oversimplified theory proposed in 1976 [110]. In addition, this approach has not yet implemented the work of fusion proteins in explicit form, taking into account possible asymmetric structures. Solving these problems will make it possible to offer complete physical models of the fusion process in various biological systems.

Generally, the modern repertoire of continuum approaches allows a detailed description of the membrane fusion, predicting and analyzing various intermediate structures and process trajectories. The possibility of constructing a continuous trajectory of the fusion process eliminated the need of postulating the geometry of fusion intermediates; the development of numerical methods eliminated the requirement of mandatory radial symmetry of these structures. In this sense, modern continuum models become a reasonable alternative to the methods of molecular dynamics. With its molecular level of detailing, the molecular dynamics method is a rather experimental approach that shows the behavior of specific systems under certain conditions, and any change in the system parameters requires performing of separate computational experiments. At the same time, the conclusions drawn in the framework of continuum models have predictive power, allowing not only interpreting the experimental results but also predicting the behavior of the entire system under the variation of the parameters.

## Figures and Tables

**Figure 1 ijms-21-03875-f001:**
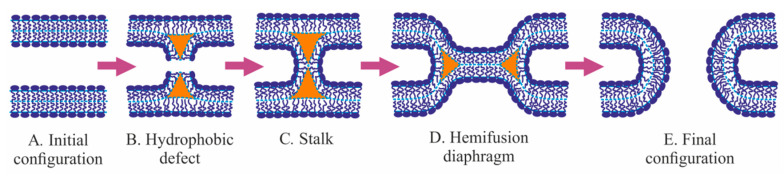
Stages of membrane fusion in the framework of the classical model. (**A**)—initial configuration of two parallel membranes; (**B**)—formation of hydrophobic defects on top of two opposing bulges on the membranes; (**C**)—a merger of contact monolayers results in the formation of the stalk; (**D**)—radial expansion of the stalk leads to hemifusion diaphragm; (**E**)—poration of the central bilayer of the hemifusion diaphragm leads to the final configuration of the fusion pore. Orange areas represent voids between contact and distal monolayers.

**Figure 2 ijms-21-03875-f002:**
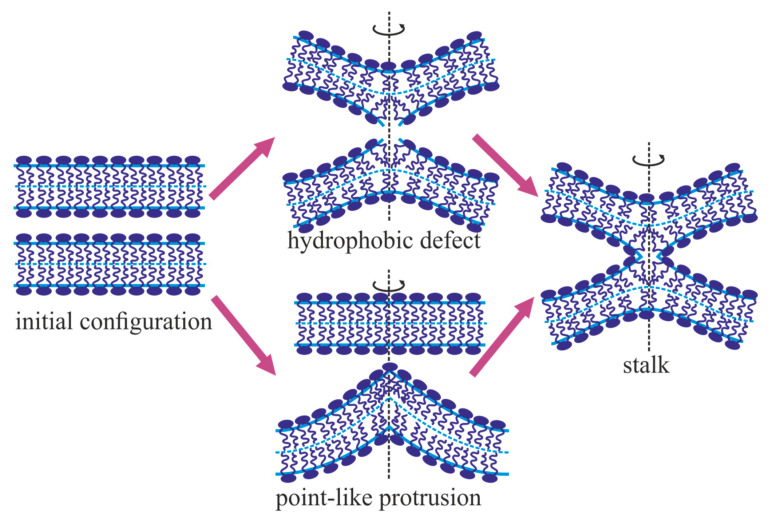
Schematic representation of membrane structures on the fusion trajectory from the initial configuration (left) to stalk (right). Two alternative intermediate states are shown: circular hydrophobic defect (above) and point-like protrusion (below).

**Figure 3 ijms-21-03875-f003:**
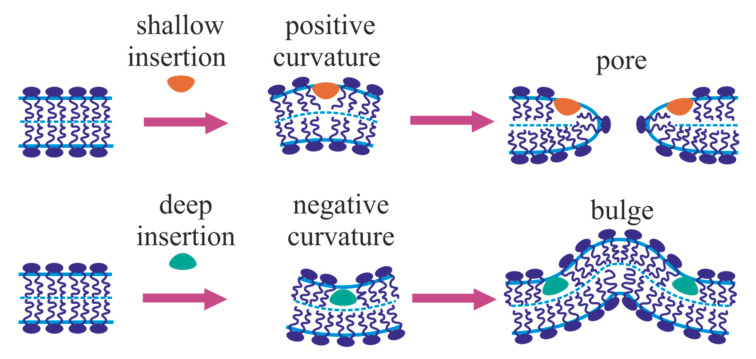
Schematic representation of the mechanism of membrane modification by the incorporated peptide. Shallow insertion (upper row) induces positive spontaneous curvature and promotes the formation of a through pore. Deep insertion (lower row) induces negative spontaneous curvature and facilitates bulge formation.

**Figure 4 ijms-21-03875-f004:**
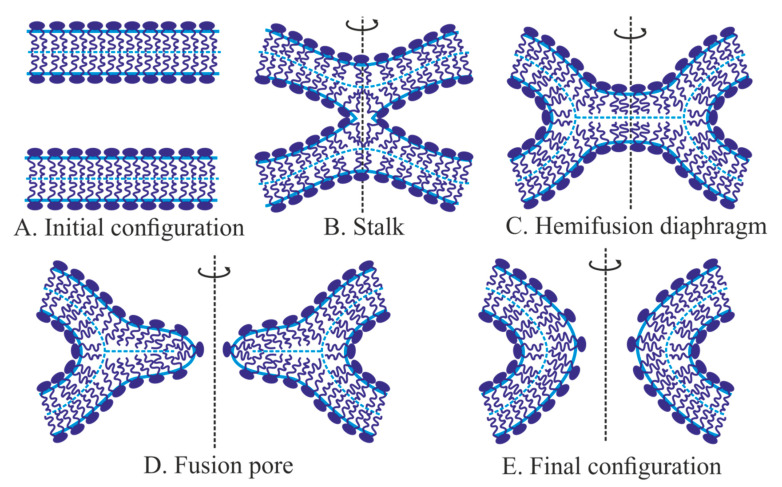
Stages of the membrane fusion process. (**A**)—Initial configuration (two locally approached bilayers); (**B**)—stalk formation; (**C**)—radial expansion of the stalk and formation of the hemifusion diaphragm; (**D**)—formation of small fusion pore; (**E**)—the accomplishment of the fusion (fusion pore expansion).

**Figure 5 ijms-21-03875-f005:**
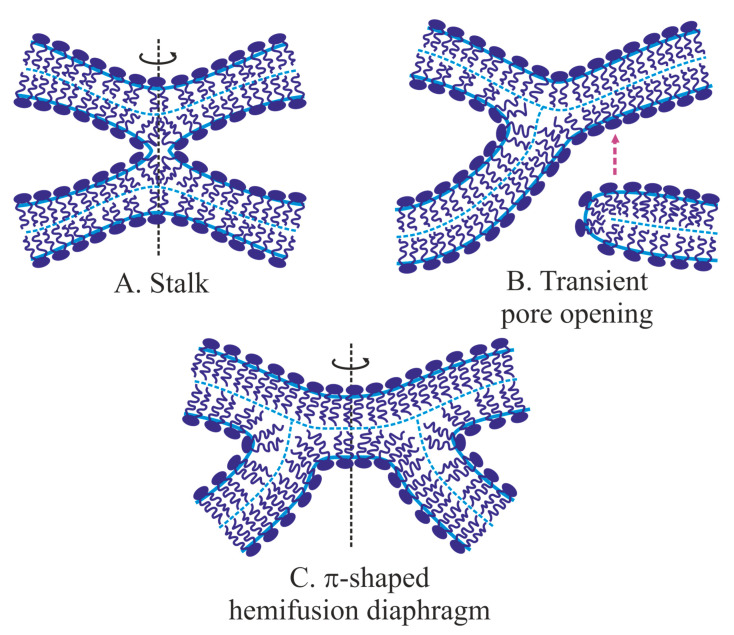
Alternative fusion path with leakage in one of the merging membranes. The following stages of the process are shown: (**A**)—stalk; (**B**)—pore formation in the vicinity of the stalk in one of the merging membranes; (**C**)—enveloping of the pore by the linearly expanding stalk results in the formation of π-shaped hemifusion diaphragm.

**Figure 6 ijms-21-03875-f006:**
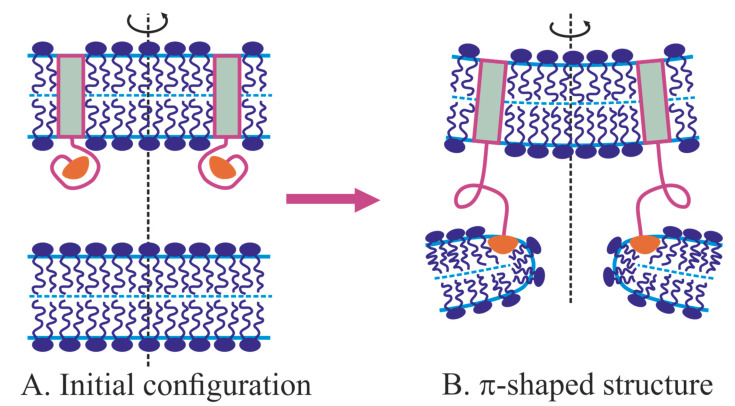
The π-shaped leaky structure, which is a dead-end pre-stalk intermediate. The model of cell infection by an influenza virus. The viral membrane is at the top, the target membrane of the cell is at the bottom. (**A**)—initial configuration; (**B**)—shallow insertion of fusion peptides leads to the formation of through pore in the target membrane of the cell. Transmembrane domains in the viral membrane are shown as grey rectangles.

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
