# Peer review of "Continuum Models of Membrane Fusion: Evolution of the Theory"

_ijms, 2020, doi:10.3390/ijms21113875_

Round 1

Reviewer 1 Report

In this manuscript, Akimov and colleagues provide a comprehensive review of continuum models of membrane fusion, emphasizing historical developments of the theoretical frameworks. The authors lay down a range of kinetics present in different biological scenarios where membrane fusion plays a vital role (viral fusion, neurotransmitter release, intracellular membrane trafficking). This is followed by a back-to-back comparison of theory development starting from early Kozlov's work in parallel with results/parameters that came out from computational methods of numerical modeling. I command the extent of the covered literature, making this a very thorough review.

One aspect to be addressed is the presence of multiple protein clusters in the membrane. These edges of these clusters have a line tension that affects the membrane remodeling. A prominent example is a hydrophobic mismatch shown both for model peptides and for several biological systems (PMID: 9805000; PMID: 25635869; PMID: 24912166).

Minor comment: 

- chapter numbering needs to be corrected: (1), (2), (5). 

Author Response

REVIEW REPORT (REVIEWER 1)

In this manuscript, Akimov and colleagues provide a comprehensive review of continuum models of membrane fusion, emphasizing historical developments of the theoretical frameworks. The authors lay down a range of kinetics present in different biological scenarios where membrane fusion plays a vital role (viral fusion, neurotransmitter release, intracellular membrane trafficking). This is followed by a back-to-back comparison of theory development starting from early Kozlov's work in parallel with results/parameters that came out from computational methods of numerical modeling. I command the extent of the covered literature, making this a very thorough review.

We thank the Reviewer for the positive evaluation.

One aspect to be addressed is the presence of multiple protein clusters in the membrane. These edges of these clusters have a line tension that affects the membrane remodeling. A prominent example is a hydrophobic mismatch shown both for model peptides and for several biological systems (PMID: 9805000; PMID: 25635869; PMID: 24912166).

We have considered mechanisms of clustering of fusion proteins driven either directly by the hydrophobic mismatch in a laterally homogeneous membrane or by lateral sorting of the proteins in a phase-separated membrane with multiple domains of different thicknesses. Possible consequences of such clustering on the fusion process are described in the text.

Lines 704-763:
[[All known fusion proteins have transmembrane domains anchored at least in one of two fusing membranes. Any TMD has a distinct length, determined by the number of successive hydrophobic amino acids. This length may differ from the thickness of the hydrophobic core of the membrane resulting in so-called hydrophobic mismatch [137]. If the membrane was flat up to the protein boundary, this would result in an exposure of either part of hydrophobic TMD or membrane hydrophobic core to the polar medium. As the contact of hydrophobic and polar media is energetically unfavorable, it is natural to assume that elastic membrane deformations should arise at the protein boundary to compensate for the hydrophobic mismatch. Deformations require mechanical work to be performed; however, the corresponding energy is lower than the energy penalty of exposure of the hydrophobic medium to the polar one. The characteristic lateral length of deformations is about units of nanometers. When two protein molecules with TMDs possessing a hydrophobic mismatch with the membrane, are far separated, deformations induced by them are independent, and elastic energy is additive. However, if proteins come closer, deformations overlap, thus leading to effective lateral interaction. The interaction is generally attractive because if two proteins come into close contact, the total length of their boundary with the membrane should decrease, reducing the elastic energy. The dependence of the energy on the distance between two transmembrane proteins has been obtained in a large number of works [138–140]. In particular, the attractive energy profile is obtained both in MD simulations [141] and in the framework of continuum model [142] for a transmembrane dimer of gramicidin A. The hydrophobic mismatch is experimentally shown to drive the organization of syntaxin 1 and syntaxin 4 (participants of SNARE complex) into clusters in laterally homogeneous membranes [143]. The TMD of syntaxin 4 is somewhat longer than the TMD of syntaxin 1. Protein clustering is observed in membranes of different thicknesses except one that
fits perfectly the length of TMD. As TMD length is different for syntaxin 1 and syntaxin 4, these proteins form non-overlapping clusters at a given thickness of the membrane [143]. Such clustering could regulate the cooperativity of mechanical forces and torques induced by fusion proteins. Besides, it may induce local membrane bending and potentially decrease the energy barrier for membrane fusion [143].

Cell membranes include hundreds of different types of lipids [144]. Under physiological conditions, lipids can undergo phase separation producing numerous microscopic domains with sizes varying in the range of 25–200 nm [145–147]. Similar domains in model membranes are more ordered than the surrounding membrane, and consequently they have higher elastic stiffness and bilayer thickness [98, 148, 149]. Besides, domains as small as tens nanometers in diameter are shown to be bilayer, i.e. to span the whole membrane [150]. It is higher elastic stiffness and larger bilayer thickness that are responsible for the bilayer structure of ordered domains [151, 152]. The larger thickness of the domain results in the hydrophobic mismatch, which compensation is possible via elastic deformations arising at the domain boundary [152]. The energy of elastic deformations is believed to be a major contribution to the domain line tension, the interphase boundary energy related to the unit length of the boundary [153]. Ordered domains represent an inhomogeneity in membrane thickness, thus allowing transmembrane proteins to choose the membrane phase (ordered or disordered), where bilayer thickness best fits the length of the protein TMD. Alternatively, transmembrane protein can induce local phase separation leading to the formation of small lipid domain of optimal bilayer thickness in its vicinity by so-called wetting mechanism [154, 155]. However, as the ordered domain possesses larger elastic moduli as compared to the surrounding membrane, the lateral distribution of transmembrane proteins is not controlled by the hydrophobic mismatch only. In the thorough analysis provided in the works [156, 157], three key determinants of targeting of transmembrane proteins to ordered domains are found: i) the length of TMD; ii) the post-translational modification of the protein (palmitoylation, miristoylation); iii) surface area of TMD side chains. The local enrichment of fusion proteins in a distinct membrane phase may regulate the cooperativity of their mechanical activity. Besides, it is demonstrated that most types of membrane inclusions, in particular, amphipathic and hydrophobic peptides, as well as transmembrane proteins with different TMD length and shape, manifest strong affinity to the boundary of the ordered domain [157, 158]. Due to the line tension of its boundary, the optimal shape of the domain is a circular one. The affinity of TMDs and/or fusion peptides to the circular domain boundary may lead to the self-organization of fusion proteins into a ring-like fusion rosette, thus allowing concerting mechanical efforts developing by fusion protein molecules [65]. Besides, if the line tension of the domain boundary is high enough, it might cause bulging (and even pinch-off) of the domain out of the membrane plane [159, 160], as this lead to a decrease of the length of the interphase boundary. Formation of bulges in merging membranes is an important stage of the fusion process; in a phase separated membrane the bulging may occur spontaneously, i.e. without consumption of the energy stored in fusion proteins.]

Minor comment:
- chapter numbering needs to be corrected: (1), (2), (5).
We have corrected the chapter numbering.

Reviewer 2 Report

This is a timely, topical review of the evolution of theoretical studies of membrane fusion from a group of investigators with abundant expertise in the field. The authors’ experience lies principally in the arena of efforts to develop realistic models of the energy landscape associated with specific fusion models and how these efforts can inform studies of everything from the fusion with cells of enveloped viruses to the exocytotic events seen at nerve terminal. It is in this effort to generalize the review’s observations that I have some concerns, which when suitably addressed by the authors, should improve this contribution. In addition, there are minor stylistic issues that I’ll enumerate, because attention to these matters should enhance the readability and impact of this paper.

  1. The introduction does a good job of pointing to the relative ubiquity of membrane fusion events. But, I’d encourage the authors to restrict their comments to eukaryotic cells, because the evidence for membrane trafficking in prokaryotic cells is very meager. Yes, there are a few examples of such processes in prokaryotic cells, but it’s still reasonable to point out that this review pertains almost exclusively to events observed in eukaryotic cells.
  2. Line 71: It would be safer to point to these estimates of the “fusion-site-volume” as residing at the low end of the spectrum. For instance, in a 2016 PNAS paper from the lab of U.J. McMahan (JH Jung et al), EM data were analyzed to obtain the average contact area between docked synaptic vesicles and the plasma membrane at motor nerve terminals. These measurements revealed that these direct contact zones greatly exceeded 100 square microns, and these results pose serious questions about how one forms a fusion stalk within such a contact area. Based on the data in the Jung paper, there simply is insufficient volume to form a stalk structure at nerve terminals. Consequently, it may be necessary to consider alternative fusion mechanisms at nerve terminals. 
  3. Line 75: The reference cited here is a poor choice and unequivocally incorrect. The first systematic studies of synaptic delay were by Katz and Miledi in the 1960s. They found that the delay between the arrival of an action potential at nerve endings and the subsequent post-synaptic response was ~0.5 msec. These observations were subsequently extended, most notably by Sabatini and Regehr in the mid-1990s. They reported that the amount of time that elapsed between the rise of presynaptic ionized Ca and the post-synaptic response was significantly less than 0.1 msec. Hence, fusion events at nerve endings happen extraordinarily rapidly, and certainly far faster than the 10s of milliseconds cited here. This kinetic alacrity is seldom addressed in fusion models and remains one of the impediments to a better understanding of fusion events as they unfold at specialized loci, such as mammalian nerve terminals. At the same time, it is worth noting here that both the Katz and Miledi studies and the Sabatini and Regehr experiments would have revealed “leakiness” if fusion events at nerve terminals were “leaky”. Since there was no hint of leakiness in their work, we can conclude that exocytosis at nerve terminals is NOT leaky. Consequently, the author’s discussion of leakiness needs to be confined to situations (viral fusion?) where there is evidence for leakiness.
  4. Line 87: The authors’ thesis here is that insights obtained by studying viral fusion may produce principles that can be applied to the understanding of other fusion events, including the extremely rapid ones that occur at nerve endings. But, I’d just like to encourage the authors to consider the alternative hypothesis: namely, that the incredible speed of fusion at nerve terminals has demanded the evolution of fusion mechanisms that are radically different from what is seen with viruses. Yes, I realize that this view contrasts with the allegations from the Rothman group concerning the similarities between viral fusion proteins and “SNAREpins”, but the fact of the matter is that all of the fusion reconstitution studies performed to date reveal fusion kinetics that are orders of magnitude SLOWER than what is observed at nerve terminals. Hence, these results can be alternatively interpreted as showing that SNAREs are inadequate as fusion machines at nerve endings and that some other process/protein (or processes/proteins) is involved. Hence, I’d encourage them to hedge here: viral fusion might be helpful, but it may also be misleading…….After all, a virus does not care how quickly it gets into a cell, as long as it gets in. If our neurons were temporally diffident, we would not be having this discussion.

  1. Comment: From Line 87 on, the authors do a reasonably thorough and systematic job of reviewing the literature that is pertinent to stalk models of membrane fusion along with forays into empirical studies that link theory and practical observation. Their efforts include helpful caveats concerning unresolved issues and comments regarding ongoing developments. This exposition includes helpful figures, an excellent conclusions section, and it all holds together well. 

6. Lines 604-608: Starting with Bangham’s work in the 1960s, many groups have studied fusion between liposomes composed of various combinations of biologically relevant lipids. A general observation from these studies is that as long as there are anionic lipids included in the lipid mix, one can induce bona fide fusion (as opposed, say, to lysis and re-annealing) by adding various fusogenic substances (including Ca ions, certain peptides, and other polymers, like PEG). Since it is not obvious how these “fusogens” can produce mechanical forces, (as is allegedly supplied by biological fusion proteins) I’m wondering whether the authors’ focus on mechanical energy barriers is too limiting. I’d encourage them to remind readers that it is the lipid part of membranes that fuse. And, that the role of proteins is analogous to the role of enzymes: they can lower the activation energy for fusion. And, proteins presumably can accelerate the forward rate constant. But, I’m unaware of any data that explicitly demonstrate how they manage these roles. Yes, folk have interpreted data from reconstitution experiments in mechanistic terms, but the fusion events in vitro operate so slowly that one certainly cannot extrapolate from in vitro results to the in vivo situation. 

Minor concerns:

Line 47: This is misleading. Synaptic vesicles fuse with the presynaptic plasma membrane and release neurotransmitter that interacts with the target cell(s). As currently written, it sounds like the vesicles are fusing with the target cell. Please revise.

Lines 43,48,54 et al.: the authors repeatedly end sentences with “etc”. This is not desirable in a scholarly communication. Either provide additional commentary/examples or simply excise the etc.

At the risk of being an ethnocentric jerk, I’d like to encourage the authors or editors to have a native English speaker “improve” the text. Although it was not a serious detriment to my perusal of the paper, this submission would benefit from solicitous editing. 

Author Response

REVIEW REPORT (REVIEWER 2)

This is a timely, topical review of the evolution of theoretical studies of membrane fusion from a group of investigators with abundant expertise in the field. The authors’ experience lies principally in the arena of efforts to develop realistic models of the energy landscape associated with specific fusion models and how these efforts can inform studies of everything from the fusion with cells of enveloped viruses to the exocytotic events seen at nerve terminal. It is in this effort to generalize the review’s observations that I have some concerns, which when suitably addressed by the authors, should improve this contribution. In addition, there are minor stylistic issues that I’ll enumerate, because attention to these matters should enhance the readability and impact of this paper.

We thank the Reviewer for the positive evaluation.

The introduction does a good job of pointing to the relative ubiquity of membrane fusion events. But, I’d encourage the authors to restrict their comments to eukaryotic cells, because the evidence for membrane trafficking in prokaryotic cells is very meager. Yes, there are a few examples of such processes in prokaryotic cells, but it’s still reasonable to point out that this review pertains almost exclusively to events observed in eukaryotic cells.

We have indicated throughout the text that the review focuses on fusion processes occurring in or involving eukaryotic cells only.

Lines 21-23:
[[In the present review, focusing on fusion processes occurring in eukaryotic cells, we scrutinize the history of these models, their evolution and complication, as well as open questions and remaining theoretical problems.]]

Lines 41-42:
[[Nevertheless, rearrangements of the membranes constantly occur in the eukaryotic cell.]]

Lines 46-47:
[[One characteristic example of the fusion in eukaryotic cells is synaptic transmission...]]

Lines 208-209:
[[In this review, we focus on the parallel evolution of the theory of elasticity and models of the fusion processes in eukaryotic cells.]]

Line 71: It would be safer to point to these estimates of the “fusion-site-volume” as residing at the low end of the spectrum. For instance, in a 2016 PNAS paper from the lab of U.J. McMahan (JH Jung et al), EM data were analyzed to obtain the average contact area between docked synaptic vesicles and the plasma membrane at motor nerve terminals. These measurements revealed that these direct contact zones greatly exceeded 100 square microns, and these results pose serious questions about how one forms a fusion stalk within such a contact area. Based on the data in the Jung paper, there simply is insufficient volume to form a stalk structure at nerve
terminals. Consequently, it may be necessary to consider alternative fusion mechanisms at nerve terminals.

We have formulated the sentence more accurately by considering separately the contact area of two merging membranes and a characteristic size of fusion intermediate structures.

Lines 73-78:
[[Experimental studies of the fusion process are difficult due to the extremely small volume of fusion intermediate structures which characteristic dimensions are approximately 10  10  10 nm3 [22, 23]. The contact area of two merging membranes may vary in a wide range, e.g. from 50-650 nm2 for the contact of a synaptic vesicle with presynaptic membrane [24] to several square microns for a fusion of two cells [25]; however, the characteristic lateral size of fusion intermediate structures is of the order of several membrane thicknesses, i.e. 10 nm [23, 25, 26].]]

Line 75: The reference cited here is a poor choice and unequivocally incorrect. The first systematic studies of synaptic delay were by Katz and Miledi in the 1960s. They found that the delay between the arrival of an action potential at nerve endings and the subsequent post-synaptic response was ~0.5 msec. These observations were subsequently extended, most notably by Sabatini and Regehr in the mid-1990s. They reported that the amount of time that elapsed between the rise of presynaptic ionized Ca and the post-synaptic response was significantly less than 0.1 msec. Hence, fusion events at nerve endings happen extraordinarily rapidly, and certainly far faster than the 10s of milliseconds cited here. This kinetic alacrity is seldom addressed in fusion models and remains one of the impediments to a better understanding of fusion events as they unfold at specialized loci, such as mammalian nerve terminals.

We have corrected the value of the characteristic time of neurotransmitter release basing on data reported in [Sabatini, B.L.; Regehr, W.G. Optical measurement of presynaptic calcium currents. Biophys. J. 1998, 74, 1549-1563; Barrett, E.F.; Stevens, C.F. The kinetics of transmitter release at the frog neuromuscular junction. J. Physiol. 1972, 227, 691-708. DOI: 10.1113/jphysiol.1972.sp010054].

Lines 79-81:
[[... the fusion during synaptic transmission occurs over times of the order of tens to hundreds of microseconds [28, 29]]]

... At the same time, it is worth noting here that both the Katz and Miledi studies and the Sabatini and Regehr experiments would have revealed “leakiness” if fusion events at nerve terminals were “leaky”. Since there was no hint of leakiness in their work, we can conclude that exocytosis at nerve terminals is NOT leaky. Consequently, the author’s discussion of leakiness needs to be confined to situations (viral fusion?) where there is evidence for leakiness.

We have explicitly indicated that the fusion of synaptic vesicles is most probably leakless, while the fusion induced by enveloped viruses may be leaky.

Lines 1033-1035:
[[The described models were developed under the assumption that membrane fusion occurs without leakage that is probably true for many fusion events. Thus, fusion in synaptic transmission seems to be completely leakless [28, 29].]]

Lines 1136-1139:
[[At the same time, experimental data were accumulated on the occurrence of leakage in the course of fusion processes in a wide variety of systems [175, 176, 186, 187]. Mention that to the best of our knowledge the leakage was never observed in the fusion of neurotransmitter-loaded vesicles with the presynaptic membrane.]]

Line 87: The authors’ thesis here is that insights obtained by studying viral fusion may produce principles that can be applied to the understanding of other fusion events, including the extremely rapid ones that occur at nerve endings. But, I’d just like to encourage the authors to consider the alternative hypothesis: namely, that the incredible speed of fusion at nerve terminals has demanded the evolution of fusion mechanisms that are radically different from what is seen with viruses. Yes, I realize that this view contrasts with the allegations from the Rothman group concerning the similarities between viral fusion proteins and “SNAREpins”, but the fact of the matter is that all of the fusion reconstitution studies performed to date reveal fusion kinetics that are orders of magnitude SLOWER than what is observed at nerve terminals. Hence, these results can be alternatively interpreted as showing that SNAREs are inadequate as fusion machines at nerve endings and that some other process/protein (or processes/proteins) is involved. Hence, I’d encourage them to hedge here: viral fusion might be helpful, but it may also be misleading…….After all, a virus does not care how quickly it gets into a cell, as long as it gets in. If our neurons were temporally diffident, we would not be having this discussion.

We have indicated particular cases where the model of viral fusion might be adequate. Besides, we explicitly manifested that the mechanism of ultrafast fusion in nerve terminals may differ qualitatively from that for the virus-induced fusion, as the quantitative characteristics of the fusion processes are very different in these cases.

Lines 89-101:
[[In this regard, virus-induced fusion, and, in particular, the fusion of influenza virions with model and cellular membranes, is currently widely used as a convenient model system for detecting patterns of membrane fusion mediated by fusion proteins [36]. Similar intermediate structures (e.g., hemifusion diaphragm) are observed for fusion processes induced by enveloped viruses and during exocytosis in secreting cells. The relatively slow fusion provided by trans-SNARE complex reconstituted in model membranes [37] seems to qualitatively resemble the viral-induced fusion. However, the analogy with viral fusion could be misleading if directly transmitted to the ultrafast fusion in vivo nerve terminals. Protein machines catalyzing fusion of enveloped viruses with target cells and synaptic vesicles with presynaptic membrane are aimed to provide very different quantitative characteristics of the fusion, first of all, different rates. Thus, there is no reason to assume similar mechanisms and trajectories of the fusion process in these two cases. Anyway, the validity of the enveloped virus model for the fusion process should be explicitly justified in each particular case.]]

Comment: From Line 87 on, the authors do a reasonably thorough and systematic job of reviewing the literature that is pertinent to stalk models of membrane fusion along with forays into empirical studies that link theory and practical observation. Their efforts include helpful caveats concerning unresolved issues and comments regarding ongoing developments. This exposition includes helpful figures, an excellent conclusions section, and it all holds together well.

We thank the Reviewer for the positive evaluation.

6. Lines 604-608: Starting with Bangham’s work in the 1960s, many groups have studied fusion between liposomes composed of various combinations of biologically relevant lipids. A general observation from these studies is that as long as there are anionic lipids included in the lipid mix, one can induce bona fide fusion (as opposed, say, to lysis and re-annealing) by adding various fusogenic substances (including Ca ions, certain peptides, and other polymers, like PEG). Since it is not obvious how these “fusogens” can produce mechanical forces, (as is allegedly supplied by biological fusion proteins) I’m wondering whether the authors’ focus on mechanical energy barriers is too limiting. I’d encourage them to remind readers that it is the lipid part of membranes that fuse. And, that the role of proteins is analogous to the role of enzymes: they can lower the activation energy for fusion. And, proteins presumably can accelerate the forward rate constant. But, I’m unaware of any data that explicitly demonstrate how they manage these roles. Yes, folk have interpreted data from reconstitution experiments in mechanistic terms, but the fusion events in vitro operate so slowly that one certainly cannot extrapolate from in vitro results to the in vivo situation.

In the previous section (2.3. Hydration and hydrophobic interactions of membranes) we indicated, that the hydration repulsion between two membranes may lead to a substantial energy barrier (even, the highest one) on the fusion trajectory. We are confident that the fusogenic substances, such as Ca2+ or polyethyleneglycol, just lower that hydration barrier. Besides, the spontaneous fusion in these model systems might mean that the hydration barrier is the highest one, and subsequent energy barriers should be lower than 40 kBT.

Lines 627-638:
[[There is much evidence for the decisive role of hydration repulsion in the fusion process. Numerous experimental data manifest that the membrane fusion could occur spontaneously if substances bridging two merging membranes (e.g., Ca2+) or dehydrating their contact (e.g., polyethyleneglycol) are added between the fusing membranes [120, 121]. These data allow concluding that bringing two membranes into close contact and overcoming the hydration repulsion are the most energy-consuming phases of the fusion process, which cannot occur spontaneously at the expense of the energy of thermal fluctuations of lipids. As the fusion in such model systems further proceeds without participation of any proteins, one could conclude that the energy barriers on the rest of the fusion trajectory should not exceed the critical height of 40 kBT [68]. Thus, the function of fusion proteins is rather analogous to the role of enzymes: they can lower the main activation energy barrier for fusion, and, probably, accelerate the forward rate constant of the process.]]

Minor concerns:
Line 47: This is misleading. Synaptic vesicles fuse with the presynaptic plasma membrane and release neurotransmitter that interacts with the target cell(s). As currently written, it sounds like the vesicles are fusing with the target cell. Please revise.

We have revised the phrase.

Lines 47-48:
[[...its key stage is the fusion of synaptic vesicles containing neurotransmitters with a presynaptic plasma membrane [6].]]

Lines 43,48,54 et al.: the authors repeatedly end sentences with “etc”. This is not desirable in a scholarly communication. Either provide additional commentary/examples or simply excise the etc.

We have excised the excessively frequent "etc."

At the risk of being an ethnocentric jerk, I’d like to encourage the authors or editors to have a native English speaker “improve” the text. Although it was not a serious detriment to my perusal of the paper, this submission would benefit from solicitous editing.
We have done our best to improve English.